# Zero-Shot Subject-Driven Video Customization with Precise Motion Control

| Subject | Motion | Generated Videos |

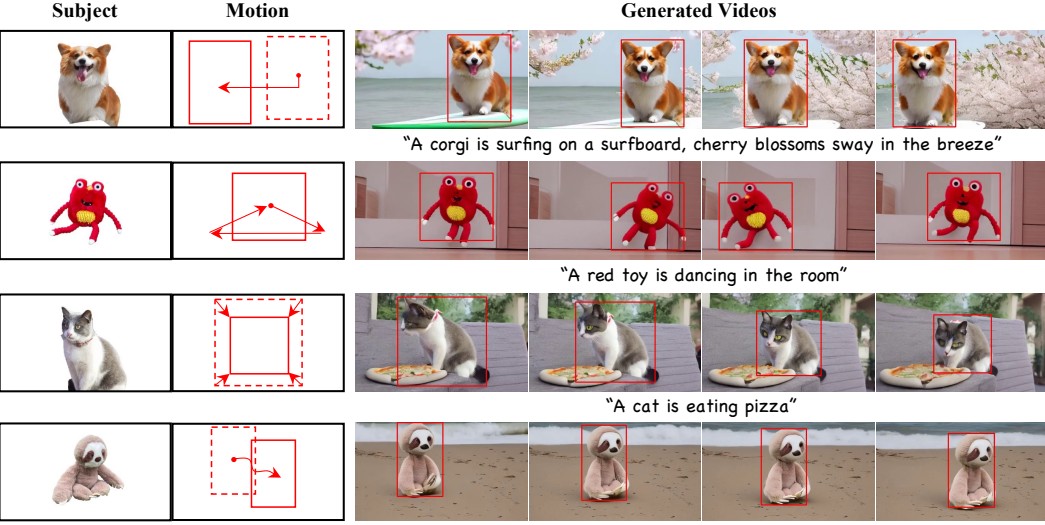

Figure 1: **Customized video generation results of DreamCustomizer**. Our method precisely generates customized subjects at specified positions **without fine-tuning at inference time**.

## ABSTRACT

Recent advances in customized video generation have enabled users to create videos tailored to both specific subjects and motion trajectories. However, existing methods often require complicated test-time fine-tuning and struggle with balancing subject learning and motion control, limiting their real-world applications. In this paper, we present **DreamCustomizer**, a zero-shot video customization framework capable of generating videos with a specific subject and motion trajectory, guided by a single image and a bounding box sequence, respectively, and without the need for test-time fine-tuning. Specifically, we introduce reference attention, which leverages the model's inherent capabilities for subject learning, and devise a mask-guided motion module to achieve precise motion control by fully utilizing the robust motion signal of box masks derived from bounding boxes. While these two components achieve their intended functions, we empirically observe that motion control tends to dominate over subject learning. To address this, we propose two key designs: **1**) the masked reference attention, which integrates a blended latent mask modeling scheme into reference attention to enhance subject representations at the desired positions, and **2**) a reweighted diffusion loss, which differentiates the contributions of regions inside and outside the bounding boxes to ensure a balance between subject and motion control. Extensive experimental results on a newly curated dataset demonstrate that DreamCustomizer outperforms state-of-the-art methods in both subject customization and motion control. The dataset, code, and models will be made publicly available.

## 1 INTRODUCTION

Customized video generation (Molad et al., 2023; Zhao et al., 2023; Wei et al., 2024) has made significant strides, largely driven by the remarkable advances in pre-trained text-to-video generation

models (Ho et al., 2022b; Wang et al., 2023a). These innovations enable users to create videos with specific subjects and precise motion trajectories (Wu et al., 2024b; Yang et al., 2024; Wang et al., 2024e), thereby broadening the scope of real-world applications for video generation.

Pioneering research efforts have explored customized video generation (Chen et al., 2023b; Jeong et al., 2024; Jiang et al., 2024; Wei et al., 2024), but they encounter significant limitations in: (1) the lack of comprehensive control over subjects and motions in a zero-shot manner, and (2) the conflict between subject learning and motion control. For instance, VideoBooth (Jiang et al., 2024) employs a tuning-free framework to inject subject embeddings from image prompts for subject customization, but it fails to control motion dynamics, leading to generated videos with minimal or absent motion. In contrast, some fine-tuning-based approaches attempt to control subject and motion simultaneously. For example, DreamVideo (Wei et al., 2024) trains two adapters separately and combines them during inference, while MotionBooth (Wu et al., 2024a) trains a customized model and manipulates attention maps to control motion during inference. However, an empirical training-inference gap persists, preventing these methods from achieving a balance between subject and motion learning. Therefore, *simultaneously enhancing and balancing subject learning and motion control in a zero-shot manner* holds great potential for practical video customization.

To that end, we propose an innovative zero-shot video customization framework, **DreamCustomizer**, which can generate videos with a specified subject and motion trajectory, derived from a *single* image and a bounding box sequence, respectively, as illustrated in Fig. 1. DreamCustomizer concurrently learns subject appearance and motion during training, allowing for harmonious subject and motion control without additional fine-tuning or manipulation during inference. To effectively inject detailed appearance information from a subject image, we introduce reference attention that leverages multi-scale features extracted from the original video diffusion model. For motion control, we devise a mask-guided motion module comprised of a spatiotemporal encoder and a spatial ControlNet (Zhang et al., 2023b), which adopts binary box masks derived from the bounding boxes as the robust motion control signal, significantly improving control precision.

While these two components can achieve their intended functions of subject and motion control, systematic experiments empirically reveal that motion control tends to dominate over subject learning, partially due to the simpler objective of generating subjects at specified positions, which compromises subject preservation quality. To mitigate this issue, we aim to strengthen the learning of subjects with two new technical contributions: **1**) the masked reference attention, which introduces a blended latent mask modeling scheme into our reference attention to enhance subject identity representations at desired positions by leveraging box masks; and **2**) a reweighted diffusion loss function, which differentiates the contributions of regions inside and outside the bounding boxes to ensure a balance between subject and motion control.

To facilitate the zero-shot video customization task, we curate a new single-subject video dataset with comprehensive annotations, comprising the caption and each frame's subject mask and bounding box. This dataset is not only larger but also considerably more diverse than previous video customization datasets. Extensive experimental results on this dataset demonstrate that DreamCustomizer outperforms state-of-the-art methods in both customization and control capabilities.

**Contributions.** The contributions of this work can be summarized as follows. **1)** We propose DreamCustomizer, the first tuning-free framework for zero-shot subject-driven video customization with precise motion trajectory control, achieved through the devised reference attention and the mask-guided motion module that uses binary box masks as motion control signals. **2)** We identify the problem of motion control dominance in DreamCustomizer, and address it by enhancing reference attention with blended masks (*i.e.*, masked reference attention) and designing a reweighted diffusion loss, effectively balancing subject learning and motion control. **3)** We curate a large, comprehensive, and diverse video dataset to support the zero-shot video customization task. Extensive experimental results demonstrate the superiority of DreamCustomizer over the existing state-of-the-art video customization methods.

## 2 RELATED WORK

**Text-to-video diffusion models.** Diffusion models have made a significant breakthrough in the generation of highly realistic samples from textual prompts (Ho et al., 2020; Rombach et al., 2022;

Podell et al., 2023). Recent advancements in text-to-video generation have expanded upon these models by incorporating temporal dynamics, enabling the production of high-quality and diverse video content (He et al., 2022; Esser et al., 2023; An et al., 2023; Zhang et al., 2023a;c; Qing et al., 2024; Wang et al., 2023c; 2024c; Singer et al., 2022; Ho et al., 2022a; Zhou et al., 2022; Wang et al., 2023d; Yuan et al., 2024; Ma et al., 2024a; Gupta et al., 2023; Bar-Tal et al., 2024; Wang et al., 2023b; Tu et al., 2024b; Xu et al., 2024a; Tu et al., 2024a; Xu et al., 2024b). VDM (Ho et al., 2022b) first introduces diffusion models into video generation by modeling the video distribution in pixel space. VLDM (Blattmann et al., 2023b) optimizes the diffusion process in the latent space to mitigate computational demands. ModelScopeT2V (Wang et al., 2023a) and VideoCrafter (Chen et al., 2023a; 2024b) incorporate spatiotemporal blocks for text-to-video generation. AnimateDiff (Guo et al., 2023b) trains a motion module appended to the pre-trained text-to-image models. SVD (Blattmann et al., 2023a) enhances the scalability of the latent video diffusion model. VideoPoet (Kondratyuk et al., 2023) investigates autoregressive video generation. Sora (Brooks et al., 2024) significantly improves the quality and stability of video generation. These advanced video generative models pave the way for customized video generation.

**Customized generation.** Customized image generation has garnered growing attention since it accommodates user preferences (Chen et al., 2023c; Han et al., 2023; Chen et al., 2024d; Wei et al., 2023; Shi et al., 2024; Li et al., 2024a; Ruiz et al., 2024; Hua et al., 2023; Han et al., 2024; Gu et al., 2024; Liu et al., 2023b; Xiao et al., 2023; Kumari et al., 2023; Liu et al., 2023c; Chen et al., 2023d). The representative works are Textual Inversion (Gal et al., 2022) and DreamBooth (Ruiz et al., 2023), where Textual Inversion optimizes text embeddings and DreamBooth fine-tunes an image diffusion model. Building upon these methods, many works explore customized video generation using a few subject or facial images (Molad et al., 2023; Chefer et al., 2024; Ma et al., 2024b; He et al., 2024). Furthermore, several works study the more challenging multi-subject video customization task (Chen et al., 2023b; Wang et al., 2024d; Chen et al., 2024c). Considering that spatial content and temporal dynamics are two indispensable components of videos, DreamVideo (Wei et al., 2024) customizes both subject and motion by training two adapters and combining them at inference time, while MotionBooth (Wu et al., 2024a) fully fine-tunes a video diffusion model to learn subjects during training and edits the attention maps to control motion during inference. However, both methods require complicated test-time fine-tuning and struggle with balancing subject and motion control due to an empirical training-inference gap. In contrast, our DreamCustomizer generates videos with harmonious subject and motion control in a tuning-free manner.

**Motion control in video generation.** Recent advancements in controllable video generation primarily focus on enhancing motion dynamics through additional control signals. Many motion customization methods learn motion patterns from intuitive reference videos (Zhao et al., 2023; Jeong et al., 2024; Ren et al., 2024; Yatim et al., 2024; Wang et al., 2024b; Wu et al., 2023), but they often require complicated fine-tuning for each motion at inference time. To circumvent the need for fine-tuning, some training-free methods manipulate attention map values through bounding boxes to control the object movements (Jain et al., 2024; Yang et al., 2024; Ma et al., 2023; Chen et al., 2024a; Qiu et al., 2024), However, these methods fail to achieve precise motion control, resulting in inconsistent frames. In contrast, several works use trajectories or coordinates as additional conditions to train a motion control module (Yin et al., 2023; Wang et al., 2024e;a; Li et al., 2024b). Nonetheless, they tend to achieve general motion control but fail to incorporate user-specified object appearances, which may limit their practical applicability. In this work, we propose masked reference attention and devise a mask-guided motion module to control the subject and motion simultaneously, effectively mitigating the control conflict using a devised reweighted diffusion loss.

# 3 PRELIMINARY

**Video diffusion models.** Video diffusion models (VDMs) (Ho et al., 2022b) aim to generate video data using diffusion processes (Ho et al., 2020). Most VDMs (Blattmann et al., 2023b; Wang et al., 2023a;b) perform the diffusion processes in a latent space using a VAE (Kingma & Welling, 2013) encoder $\mathcal{E}$ to map a video $\boldsymbol{x}_0 \in \mathbb{R}^{F \times H \times W \times 3}$ into its latent code $\boldsymbol{z}_0 = \mathcal{E}(\boldsymbol{x}_0)$, $\boldsymbol{z}_0 \in \mathbb{R}^{F \times h \times w \times 4}$, and a decoder $\mathcal{D}$ to reconstruct the video $\hat{\boldsymbol{x}}_0 = \mathcal{D}(\boldsymbol{z}_0)$. The forward process gradually adds noise to the latent code $\boldsymbol{z}_0$ according to a predetermined schedule $\{\beta_t\}_{t=1}^T$ with $T$ steps: $\boldsymbol{z}_t = \sqrt{\bar{\alpha}_t}\boldsymbol{z} + \sqrt{1 - \bar{\alpha}_t}\epsilon$, where $\bar{\alpha}_t = \prod_{s=1}^t \alpha_s$, $\alpha_t = 1 - \beta_t$, and $\epsilon \in \mathcal{N}(0, 1)$ is random noise from a Gaussian distribution.

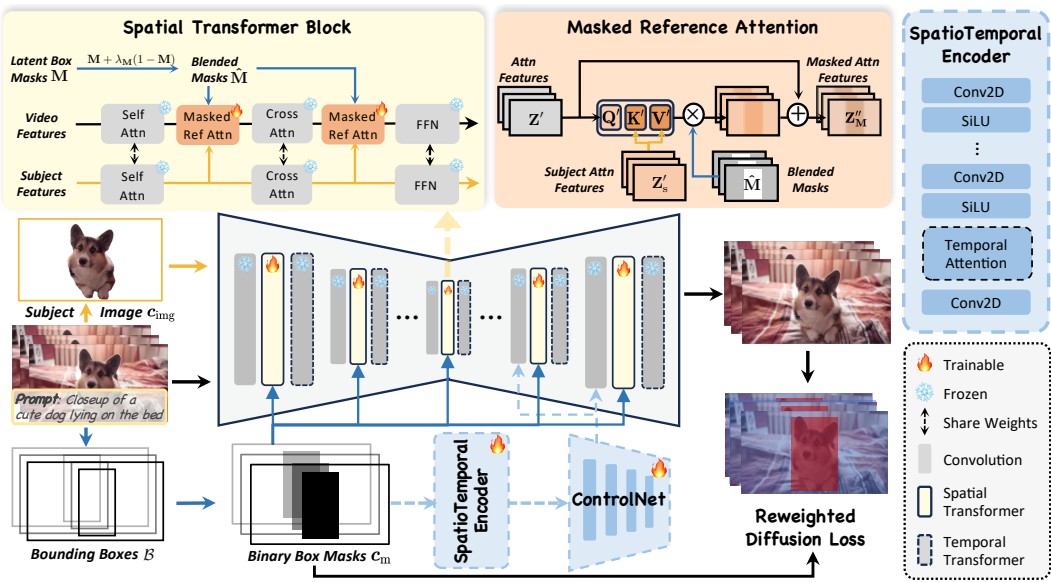

Figure 2: **Overall framework of DreamCustomizer**. During training, a random video frame is segmented to obtain the subject image with a blank background. The bounding boxes extracted from the training video are converted into binary box masks. Then, the subject image is treated as a single-frame video and processed in parallel with the video by masked reference attention that incorporates blended masks to learn the subject appearance. Meanwhile, box masks are fed into a motion module that includes a spatiotemporal encoder and a ControlNet for motion control. Both the masked reference attention and motion module are trained using a reweighted diffusion loss.

The reverse process adopts a network $\epsilon_\theta$ to predict the added noise $\epsilon$ at each timestep $t$ based on an additional condition $c$. The training objective can be simplified as a reconstruction loss:

$$\mathcal{L}(\theta) = \mathbb{E}_{z,\epsilon,c,t} \left[ \| \epsilon - \epsilon_\theta(z_t, c, t) \|_2^2 \right]. \tag{1}$$

**Attention mechanism in VDMs.** In most text-to-video VDMs, self-attention serves to capture contextual features, while cross-attention facilitates the integration of additional conditions, such as textual features $c_{\text{txt}}$. Given the features $\mathbf{Z}$ from the latent code, the standard formulation of the attention mechanism can be expressed as:

$$\mathbf{Z}' = \text{Attention}(\mathbf{Q}, \mathbf{K}, \mathbf{V}) = \text{Softmax}\left( \frac{\mathbf{Q}\mathbf{K}^\top}{\sqrt{d}} \right) \mathbf{V}, \tag{2}$$

where $\mathbf{Z}'$ is the output attention features. $\mathbf{Q}$, $\mathbf{K}$, and $\mathbf{V}$ are the query, key, and value matrices, respectively. For self-attention, $\mathbf{Q} = \mathbf{Z}\mathbf{W}_Q$, $\mathbf{K} = \mathbf{Z}\mathbf{W}_K$, $\mathbf{V} = \mathbf{Z}\mathbf{W}_V$, and for cross-attention, $\mathbf{Q} = \mathbf{Z}\mathbf{W}_Q$, $\mathbf{K} = c\mathbf{W}_K$, $\mathbf{V} = c\mathbf{W}_V$. Here, $\mathbf{W}_Q$, $\mathbf{W}_K$, $\mathbf{W}_V$ are the corresponding projection matrices. $d$ is the dimension of key features.

## 4 METHODOLOGY

Given a single subject image that defines the subject's appearance and a bounding box sequence that delineates the motion trajectory, our **DreamCustomizer** aims to generate videos featuring specified subjects and motion trajectories without fine-tuning or manipulation at inference time, as illustrated in Fig. 2. To learn the subject appearance, we leverage the model's inherent capabilities and introduce reference attention in Sec. 4.1. For motion control, we propose using box masks as the motion control signal and devise a mask-guided motion module in Sec. 4.2. Furthermore, to balance subject learning and motion control, we enhance reference attention with blended masks (*i.e.*, masked reference attention) and design a reweighted diffusion loss in Sec. 4.3. Finally, we detail the training, inference, and dataset construction processes in Sec. 4.4.

### 4.1 Subject Learning via Reference Attention

For subject learning, we focus on using a single image to capture the appearance details, which is challenging but facilitates real-world applications. Given a single input image, we first segment it to obtain the subject image $\mathbf{c}_{\text{img}}$ with a blank background, effectively preserving distinct identity features while minimizing background interference (Chen et al., 2024e; Jiang et al., 2024).

To capture the intricate details of the subject's appearance, previous works usually employ an extra image encoder (*e.g.*, CLIP (Ye et al., 2023; Jiang et al., 2024), ControlNet-like encoder (Chen et al., 2023d), ReferenceNet (Hu, 2024)) to extract image features. However, incorporating additional networks tends to escalate both parameter counts and training costs. In this work, we identify that the video diffusion model itself is capable of extracting appearance features, thus improving training efficiency without requiring auxiliary modules.

To that end, we introduce reference attention, which leverages the model's inherent capabilities to extract multi-scale subject features. Specifically, we treat the subject image as a single-frame video and input it into the original video diffusion model to obtain subject attention features $\mathbf{Z}'_s$, which is the output of self-attention or cross-attention according to Eq. (2). Our reference attention infuses the subject attention features into video attention features $\mathbf{Z}'$ by implementing a residual cross-attention:

$$\mathbf{Z}'' = \mathbf{Z}' + \text{Attention}(\mathbf{Q}', \mathbf{K}', \mathbf{V}'), \tag{3}$$

where $\mathbf{Q}' = \mathbf{Z}'\mathbf{W}'_Q$, $\mathbf{K}' = \mathbf{Z}'_s\mathbf{W}'_K$, $\mathbf{V}' = \mathbf{Z}'_s\mathbf{W}'_V$. $\mathbf{W}'_Q$, $\mathbf{W}'_K$, and $\mathbf{W}'_V$ are the projection matrices of reference attention and are initialized randomly. In addition, we initialize the weights of the output linear layer in reference attention with zeros to protect the pre-trained model from being damaged at the beginning of training (Zhang et al., 2023b; Wei et al., 2024).

### 4.2 Motion Control via Mask-Guided Motion Module

To facilitate motion control, we utilize bounding boxes as user inputs to delineate subject trajectories, offering both flexibility and convenience. We define an input sequence of bounding boxes as $\mathcal{B} = [\mathcal{B}_1, \mathcal{B}_2, \ldots, \mathcal{B}_F]$, where each box $\mathcal{B}_i$ includes coordinates of its top-left and bottom-right corners. Then, we convert these bounding boxes into a binary box mask sequence $\mathcal{M} = [\mathcal{M}_1, \mathcal{M}_2, \ldots, \mathcal{M}_F]$, where each mask $\mathcal{M}_i \in \mathbb{R}^{H \times W}$ has pixel values of 1 for the foreground subject and 0 for the background.

The final motion control signal is represented as $\boldsymbol{c}_{\text{m}} = 1 - \mathcal{M}$ to align with the subject image containing a blank background. Compared to directly using trajectories for training in previous work (Wang et al., 2024e), the box masks provide enhanced control signals and constrain subjects within the bounding box, improving training efficiency and motion control precision.

To capture motion information from the box mask sequence, we devise a mask-guided motion module, which employs a spatiotemporal encoder and a spatial ControlNet (Zhang et al., 2023b), as depicted in Fig. 2. While previous research (Guo et al., 2023a) demonstrates the efficacy of a 3D ControlNet for extracting control information from sequential inputs, its high training costs present potential drawbacks in practical applications. Given the straightforward temporal relationships in the box mask sequence, we establish that a lightweight spatiotemporal encoder is adequate for extracting the necessary temporal information. Thus, we only employ a spatial ControlNet appended to this encoder to further enhance control precision. The spatiotemporal encoder consists of repeated 2D convolutions and non-linear layers, followed by two temporal attention layers and an output convolutional layer, as shown in the right side of Fig. 2. In addition, the spatial ControlNet extracts multi-scale features and adds them to the input of convolutional layers of the VDM's decoder blocks.

### 4.3 Balancing Subject Learning and Motion Control

While the above two components achieve their intended functions, we empirically observe that motion control tends to dominate over subject learning, which compromises identity preservation quality. As shown in Fig. 3(b), the model learns motion control using a few steps, partially due to the simpler objective of generating subjects at specified positions. In Fig. 3(c), joint training of the reference attention and motion module retains the dominance of motion control, even with extended training steps, resulting in corrupted subject identity. In contrast, as shown in Fig. 3(d), our method effectively balances subject learning and motion control by proposing the following two key designs.

**Masked reference attention.** To enhance the subject identity representations at desired positions, we introduce blended latent mask modeling into our reference attention through binary box masks. Specifically, we resize the binary box masks $\mathcal{M}$ into latent box masks $\mathbf{M} = [\mathbf{M}_1, \mathbf{M}_2, \dots, \mathbf{M}_F | \mathbf{M}_i \in \mathbb{R}^{h \times w}]$ to match the size of attention features across different layers.

Then, we assign a relatively lower weight to the background (*i.e.*, regions outside the bounding boxes) in $\mathbf{M}$ to obtain blended masks $\hat{\mathbf{M}}$, forcing the model to focus more on the subject and less on the background at the feature level:

$$\hat{\mathbf{M}} = \mathbf{M} + \lambda_{\mathbf{M}}(1 - \mathbf{M}), \qquad (4)$$

where $\lambda_{\mathbf{M}}$ is the weight of background in mask. Compared to using binary masks $\mathbf{M}$, which ignore background information, blended masks $\hat{\mathbf{M}}$ can enhance the subject representations at desired positions while mitigating the background distortion. Finally, our masked reference attention can be formulated as:

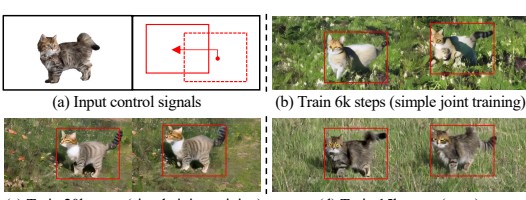

(a) Input control signals     (b) Train 6k steps (simple joint training)

(c) Train 20k steps (simple joint training)     (d) Train 15k steps (**ours**)

Figure 3: **Illustration of motion control domination in DreamCustomizer**. As seen in (b) and (c), motion control tends to dominate over subject learning during training, causing the degradation of subject identity. In (d), our method ensures a balance between subject and motion control.

$$\mathbf{Z}''_{\mathbf{M}} = \mathbf{Z}' + \hat{\mathbf{M}} \cdot \mathrm{Attention}(\mathbf{Q}', \mathbf{K}', \mathbf{V}'), \qquad (5)$$

where $\cdot$ denotes the element-wise multiplication operation. For subject learning, we freeze all original UNet parameters and only train the masked reference attentions, which are appended to both self-attention and cross-attention within each spatial transformer block, as shown in Fig. 2.

**Reweighted diffusion loss.** To balance subject learning and motion control, we further propose a reweighted diffusion loss that differentiates the contributions of regions inside and outside the bounding boxes to the standard diffusion loss. Specifically, we amplify the contributions within bounding boxes to enhance subject learning while preserving the original diffusion loss for regions outside these boxes. Our designed reweighted diffusion loss can be defined as:

$$\mathcal{L}(\theta) = \mathbb{E}_{\boldsymbol{z}, \epsilon, \boldsymbol{c}, t}\left[ \left( \underbrace{\lambda_{\mathcal{L}}\mathbf{M}}_{\text{inside}} + \underbrace{(1 - \mathbf{M})}_{\text{outside}} \right) \cdot \left\| \epsilon - \epsilon_\theta(\boldsymbol{z}_t, \boldsymbol{c}_{\text{txt}}, \boldsymbol{c}_{\text{img}}, \boldsymbol{c}_{\text{m}}, t) \right\|_2^2 \right], \qquad (6)$$

where $\lambda_{\mathcal{L}} > 1$ is the loss weight to adjust the subject identity enhancement.

### 4.4 TRAINING, INFERENCE, AND DATASET CONSTRUCTION

**Training.** We randomly select a frame from the training video and segment it to obtain the subject image with a blank background, which alleviates overfitting compared to using the first frame as in (Jiang et al., 2024). We also extract the subject's bounding boxes from all frames of the training video and convert them into box masks as the motion control signal. During training, we freeze the original 3D UNet parameters and jointly train the newly added masked reference attention, spatiotemporal encoder, and ControlNet according to Eq. (6).

**Inference.** Our DreamCustomizer is tuning-free and does not require attention map manipulations during inference. Users only need to provide a subject image and a bounding box sequence to flexibly generate customized videos featuring the specified subject and motion trajectory. The bounding boxes can be derived from various types of signals, including boxes of the first and last frames, a bounding box of the first frame accompanied by a motion trajectory, or a reference video. These signals are then converted into binary box masks for input.

**Dataset Construction.** To facilitate the zero-shot video customization task with subject and motion control, we curate a single-subject video dataset containing both video masks and bounding boxes from the WebVid-10M (Bain et al., 2021) dataset and our internal data. Annotations are generated using the Grounding DINO (Liu et al., 2023a), SAM (Kirillov et al., 2023), and DEVA (Cheng et al., 2023) models. The comparison of our dataset and previous datasets is presented in Tab. 1. Currently, we have processed 261,118 videos for training, and more details are in Appendix A.1.

| | Number of Videos | Number of Object Classes | Caption | Mask of All Frames | Box of All Frames |
|---|---|---|---|---|---|
| WebVid-10M (Bain et al., 2021) | ~10M | - | ✓ | ✗ | ✗ |
| UCF-101 (Soomro et al., 2012) | 13,320 | - | ✗ | ✗ | ✗ |
| DAVIS (Pont-Tuset et al., 2017) | 50 | 50 | ✗ | ✓ | ✓ |
| GOT-10k (Huang et al., 2019) | 9,695 | 563 | ✗ | ✗ | ✓ |
| VideoBooth Dataset (Jiang et al., 2024) | 48,724 | 9 | ✓ | ✗ | ✗ |
| **DreamCustomizer Dataset** | **261,118** | **8,197** | ✓ | ✓ | ✓ |

Table 1: **Comparsion of our dataset with related video datasets.** Our dataset contains comprehensive annotations, and is larger and more diverse than previous video customization datasets.

## 5 EXPERIMENT

### 5.1 EXPERIMENTAL SETUP

**Datasets.** We train DreamCustomizer on our curated video dataset and evaluate it through a collected test set containing 50 subjects, 36 bounding boxes, and 60 text prompt templates. The subject images are sourced from previous papers (Ruiz et al., 2023; Kumari et al., 2023) and the Internet, while bounding boxes are obtained from the videos in DAVIS dataset (Pont-Tuset et al., 2017) and boxes used in FreeTraj (Qiu et al., 2024); see Appendix A.2 for more details on experimental setting.

**Implementation details.** We jointly train all modules using the AdamW (Loshchilov, 2017) optimizer with a learning rate of 1e-4. The weight decay is set to 0, and the training iteration is 30,000. We set blended mask weight $\lambda_M$ to 0.75 and reweighted diffusion loss weight $\lambda_{\mathcal{L}}$ to 2 for training. The spatial resolution of the videos is 448×256, and the number of video frames $F$ is 16. We set the total batch size to 144, and adopt ModelScopeT2V (Wang et al., 2023a) as the base model. During inference, we employ 50-step DDIM (Song et al., 2020) and classifier-free guidance (Ho & Salimans, 2022) with guidance scale 9.0 to generate 8-fps videos.

**Baselines.** We compare our method with DreamVideo (Wei et al., 2024) and MotionBooth (Wu et al., 2024a) for both subject customization and motion control. We also compare with DreamVideo and VideoBooth (Jiang et al., 2024) for independent subject customization, while Peekaboo (Jain et al., 2024), Direct-a-Video (Yang et al., 2024), and MotionCtrl (Wang et al., 2024e) for motion trajectory control. More implementation details of all methods are provided in Appendix A.2.

**Evaluation metrics.** We evaluate our method using 9 metrics, focusing on three aspects: overall consistency, subject fidelity, and motion control precision. **1)** For overall consistency, we employ CLIP image-text similarity (CLIP-T), Temporal Consistency (T. Cons.) (Esser et al., 2023), and Dynamic Degree (DD) (Huang et al., 2024) metrics. DD uses optical flow to measure motion dynamics. **2)** For subject fidelity, we introduce four metrics: CLIP image similarity (CLIP-I), DINO image similarity (DINO-I), region CLIP-I (R-CLIP), and region DINO-I (R-DINO) metrics (Ruiz et al., 2023; Wei et al., 2024; Wu et al., 2024a). R-CLIP and R-DINO compute the similarities between the subject image and frame regions defined by bounding boxes, following (Wu et al., 2024a). **3)** For motion control precision, we use the Mean Intersection of Union (mIoU) and Centroid Distance (CD) metrics (Qiu et al., 2024). CD computes the normalized distance between the centroid of the generated subject and target bounding boxes. We use Grounding-DINO (Liu et al., 2023a) to predict the bounding boxes of generated videos. More details of metrics are reported in Appendix A.2.

### 5.2 MAIN RESULTS

**Joint subject customization and motion control.** We conduct qualitative comparison between our method and baselines for generating videos featuring both specified subjects and motion trajectories, as depicted in Fig. 4. We observe that DreamVideo and MotionBooth struggle with balancing subject preservation and motion control, especially when trained on a single subject image. We argue that the imbalanced control strengths of subject and motion hinder their performance, leading to trade-offs where enhancing one aspect degrades another. In contrast, our DreamCustomizer harmoniously generates customized videos with desired subject appearances and motion movements under various contexts. Furthermore, our method effectively constrains subjects within the bounding boxes, better aligning with user preferences and improving real-world applicability.

Figure 4: **Qualitative comparison of joint subject customization and motion control**. Dream-Customizer generates videos with customized subjects and precise motion trajectory control, while other methods suffer from the control conflict, especially when trained on a single subject image.

| Method | CLIP-T | R-CLIP | R-DINO | CLIP-I | DINO-I | T. Cons. | mIoU | CD $\downarrow$ |
|---|---|---|---|---|---|---|---|---|
| DreamVideo | 0.289 | 0.682 | 0.244 | 0.692 | 0.386 | 0.966 | 0.169 | 0.196 |
| MotionBooth | 0.267 | 0.708 | 0.301 | 0.686 | 0.383 | **0.970** | 0.351 | 0.097 |
| **DreamCustomizer** | **0.303** | **0.751** | **0.392** | **0.694** | **0.411** | 0.968 | **0.670** | **0.048** |

Table 2: **Quantitative comparison of joint subject customization and motion control.**

The quantitative comparison results are presented in Tab. 2. Our DreamCustomizer consistently surpasses all baseline methods in text alignment, subject fidelity, and motion control precision, while achieving comparable Temporal Consistency. Notably, our approach significantly outperforms the baselines in the mIoU and CD metrics, verifying our robust motion control capabilities. In contrast, DreamVideo shows the second-best CLIP-I and DINO-I scores but inferior mIoU and CD, indicating its strength in preserving subject identity despite limitations in motion movements. MotionBooth exhibits the lowest CLIP-T due to the fine-tuning of the whole model, but achieves better mIoU and CD metrics than DreamVideo, suggesting that using explicit motion control signals (*e.g.*, bounding boxes) may be more effective than learning from the reference video.

**Subject customization.** We evaluate the independent subject customization capabilities. Fig. 5 presents qualitative comparison results. We observe that VideoBooth exhibits limited generalization for subjects not included in its training data, while DreamVideo fails to capture appearance details when trained on a single image. In contrast, when trained on the same dataset as VideoBooth, our DreamCustomizer with reference attention and reweighted diffusion loss generates videos with desired subjects while conforming to textual prompts.

| Method | CLIP-T | CLIP-I | DINO-I | T. Cons. | DD |
|---|---|---|---|---|---|
| DreamVideo | 0.290 | 0.714 | 0.470 | **0.975** | 0.592 |
| VideoBooth | 0.274 | **0.724** | 0.459 | 0.970 | 0.780 |
| **DreamCustomizer** | **0.297** | 0.721 | **0.472** | 0.972 | **0.952** |

Table 3: **Quantitative comparison of subject customization.**

Tab. 3 shows the quantitative comparison results. While DreamCustomizer remains comparable CLIP-I and Temporal Consistency, it achieves the highest CLIP-T, DINO-I, and Dynamic Degree, verifying the superior of our method in text alignment, subject fidelity, and motion dynamics.

**Motion control.** Besides subject customization, we also evaluate the motion control capabilities, as shown in Fig. 6. The results suggest that all baselines struggle to accurately control subject movements as defined by bounding boxes. Meanwhile, Direct-a-Video may generate videos with corrupted object appearances due to its manipulation of attention map values.

| Method | CLIP-T | T. Cons. | mIoU | CD $\downarrow$ |
|---|---|---|---|---|
| Peekaboo | 0.318 | 0.968 | 0.322 | 0.117 |
| Direct-a-Video | 0.312 | 0.965 | 0.355 | 0.124 |
| MotionCtrl | 0.321 | **0.971** | 0.248 | 0.122 |
| **DreamCustomizer** | **0.322** | 0.969 | **0.752** | **0.039** |

Table 4: **Quantitative comparison of motion control.**

In contrast, DreamCustomizer with only motion encoder achieves precise motion control and effectively ensures subjects remain within the bounding boxes, demonstrating robust control capabilities.

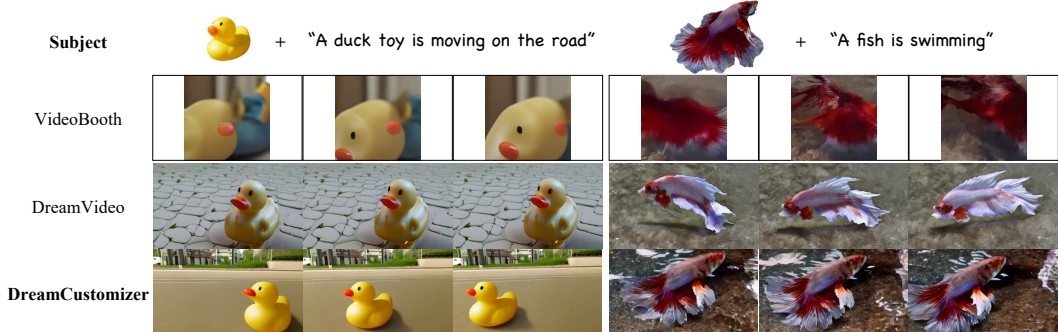

Figure 5: **Qualitative comparison of subject customization**. DreamCustomizer generates videos with accurate subject appearance and enhanced motion dynamics, aligning with provided prompts.

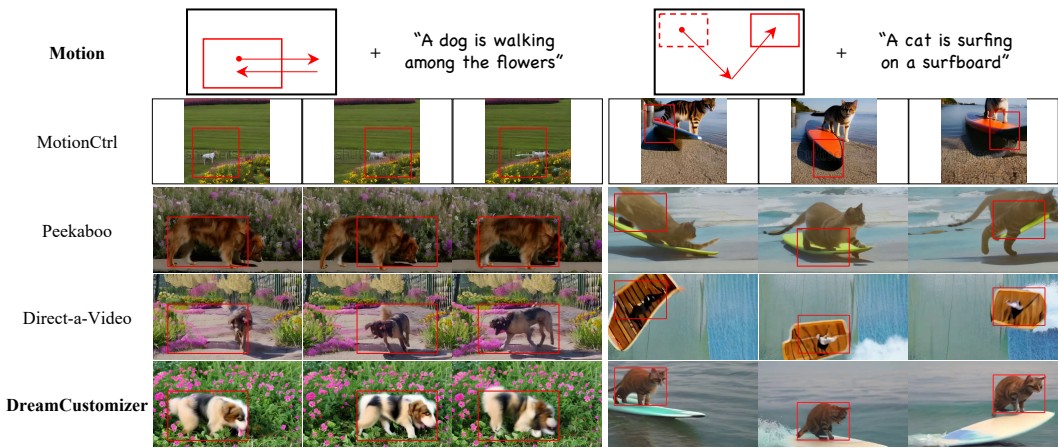

Figure 6: **Qualitative comparison of motion control**. Our DreamCustomizer achieves precise motion trajectory control and effectively maintains subjects within the specified bounding boxes.

As shown in Tab. 4, our method, while exhibiting a slightly lower T. Cons. compared to MotionCtrl, achieves the highest CLIP-T and substantially outperforms baselines in both mIoU and CD metrics.

**User study.** We conduct user studies to further evaluate our DreamCustomizer. We ask 15 annotators to rate 300 groups of videos generated by three methods. Each group contains 3 generated videos, a subject image, a textual prompt, and corresponding bounding boxes. We evaluate all methods with a majority vote from four aspects: Text Alignment, Subject Fidelity, Motion Alignment, and Overall Quality. Results in Fig. 7 indicate that our method is most preferred by users across four aspects; see Appendix A.4 for more details of user study.

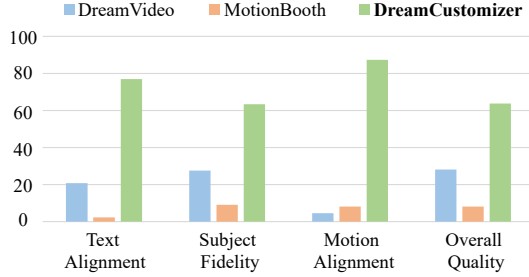

Figure 7: **Human evaluation** on joint subject customization and motion control.

### 5.3 ABLATION STUDIES

**Effects of each component.** We perform an ablation study on the effects of each component, as shown in Fig. 8(a). We observe that without the mask mechanism or the reweighted diffusion loss, the quality of subject identity degrades due to the dominance of motion control. While employing binary masks in masked reference attention helps retain subject identity, it often results in a blurry

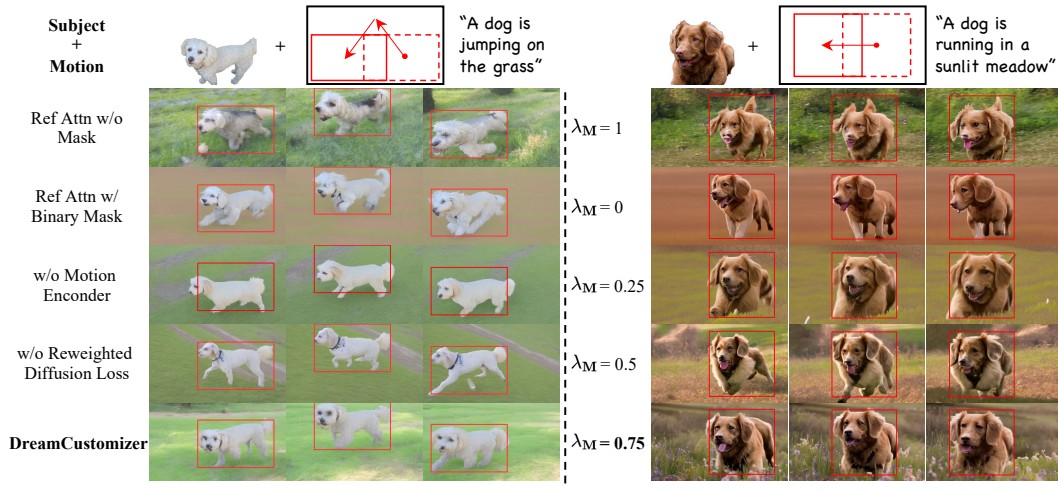

Figure 8: **Qualitative ablation studies** on each component and blended mask weight.

| | CLIP-T | R-CLIP | R-DINO | CLIP-I | DINO-I | T. Cons. | mIoU | CD↓ |
|---|---|---|---|---|---|---|---|---|
| Ref Attn w/o Mask ($\lambda_\mathrm{M} = 1$) | 0.301 | 0.744 | 0.370 | 0.682 | 0.375 | 0.963 | 0.601 | 0.055 |
| Ref Attn w/ Binary Mask ($\lambda_\mathrm{M} = 0$) | 0.293 | **0.755** | 0.388 | **0.696** | 0.394 | 0.967 | **0.706** | 0.044 |
| Ref Attn w/ Blended Mask ($\lambda_\mathrm{M} = 0.25$) | 0.299 | 0.748 | 0.379 | 0.685 | 0.395 | 0.964 | 0.693 | **0.041** |
| Ref Attn w/ Blended Mask ($\lambda_\mathrm{M} = 0.5$) | 0.301 | 0.748 | 0.376 | 0.694 | 0.386 | 0.961 | 0.664 | 0.051 |
| w/o Motion Encoder | 0.302 | 0.731 | 0.325 | 0.690 | 0.389 | 0.963 | 0.587 | 0.062 |
| w/o Reweighted Diffusion Loss | 0.300 | 0.740 | 0.362 | 0.673 | 0.382 | 0.961 | 0.650 | 0.053 |
| **DreamCustomizer** ($\lambda_\mathrm{M} = 0.75$) | **0.303** | 0.751 | **0.392** | 0.694 | **0.411** | **0.968** | 0.670 | 0.048 |

Table 5: **Quantitative ablation studies** on each component and blended mask weight.

background and low-quality video due to ignoring the background information in attention. Notably, without the motion encoder, our masked reference attention still achieves rough trajectory control.

Quantitative results in Tab. 5 demonstrate that removing the mask mechanism, motion encoder, or reweighted diffusion loss consistently degrades performance across all metrics. This confirms that each component contributes to the overall performance; see Appendix A.3 for more ablation studies.

**Effects of blended mask weight $\lambda_\mathrm{M}$.** To determine the optimal blended mask weight $\lambda_\mathrm{M}$, we vary its value and measure its impact. As shown in Fig. 8(b), using $\lambda_\mathrm{M} = 1$ results in a degradation of subject identity, while $\lambda_\mathrm{M} = 0$ leads to blurred backgrounds. We also observe that increasing $\lambda_\mathrm{M}$ can enhance video quality. To balance subject identity and video quality, we finalize on $\lambda_\mathrm{M} = 0.75$.

Tab. 5 shows the quantitative results. $\lambda_\mathrm{M} = 0$ causes the worst CLIP-T but the highest mIoU. We argue that a smaller $\lambda_\mathrm{M}$ enhances positional information but suppresses background, resulting in improved control precision but degraded video quality. Additionally, results indicate that using blended masks consistently outperforms its absence in subject fidelity, underscoring its efficacy.

# 6 CONCLUSION

In this paper, we present DreamCustomizer, a novel zero-shot video customization framework that generates videos with specified subjects and motion trajectories. We introduce reference attention for subject learning and devise a mask-guided motion module for motion control. To address the problem of motion control dominance in DreamCustomizer, we introduce blended masks into reference attention and design a reweighted diffusion loss, effectively balancing subject learning and motion control. Extensive experimental results on our newly curated video dataset demonstrate the superiority of DreamCustomizer in both subject customization and motion trajectory control.

**Limitations.** Although our method can customize a single subject with a single trajectory, it fails to generate videos containing multiple subjects and trajectories. One solution is to construct a more diverse dataset and train a general model. We provide more discussions in Appendix A.5.

## 7 ETHICS STATEMENT

Unlike previous video customization methods that require complicated test-time fine-tuning, our approach enables users to flexibly create customized videos featuring specified subjects and motion trajectories, without the need for fine-tuning or manipulation during inference. This tuning-free paradigm significantly enhances the real-world applications of customized video generation. Nonetheless, our method still encounters challenges common to generative models, such as the potential for creating fake data. Implementing robust video forgery detection techniques may address these concerns. In addition, we commit to adhering to ethical guidelines when releasing our dataset.

## 8 REPRODUCIBILITY STATEMENT

We make the following efforts to ensure the reproducibility of DreamCustomizer: (1) Our dataset, code, and trained model weights will be made publicly available. (2) We provide the complete descriptions of the dataset construction pipeline in Appendix A.1. (3) We provide implementation details in Sec. 5.1 and Appendix A.2. (4) We present the details of the human evaluation setups in Appendix A.4.

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

# A APPENDIX

## A.1 DATASET CONSTRUCTION

To facilitate the task of zero-shot video customization with subject and motion control, we curate a single-subject video dataset that encompasses video captions, video masks, and bounding boxes from the WebVid-10M (Bain et al., 2021) dataset and our internal data. The WebVid-10M dataset comprises 10 million video-text data pairs and is widely used for text-to-video generation.

We obtain comprehensive annotations by segmenting the subjects of all frames for each video using the Grounding DINO (Liu et al., 2023a), SAM (Kirillov et al., 2023), and DEVA (Cheng et al., 2023) models, as shown in Fig. 9. Specifically, we first extract noun chunks as the initial subject word from the video caption using the spaCy and NLTK library. For videos that lack the caption, we use a pretrained Visual Language Model (Lin et al., 2024) to get its textual description. Then, we use the NLTK library to perform lemmatization and filter out non-words while asking some annotators to refine the subject words to better align with the video content. Subsequently, we generate the first frame's bounding boxes using Grounding DINO based on the subject word and feed the bounding boxes into SAM to get the subject mask. We then utilize the object tracker DEVA to populate the mask across all frames of the video, thereby acquiring bounding boxes and masks for all frames.

Since we focus on single-subject video customization, we filter out videos that contain multiple subjects for the subject word by the number of bounding boxes in the first frame. We also filter out subjects that are either too small or too large (*i.e.*, those nearly matching the size of the entire video) by assessing the ratio of the width, height, and area of the subject's bounding box to the entire video. To improve the annotation precision, we set a relatively high threshold to filter out detections that the model is uncertain about. Furthermore, we observe a considerable proportion of WebVid-10M videos lacking substantial subject movements. To ensure the motion dynamic of our dataset, we evaluate each video in the WebVid-10M dataset by comparing their bounding boxes of the first and last frames, retaining those clips where sufficient differences exist between these frames.

After data filtering, we obtain 261,118 video data pairs and 8,197 subject classes in the current version. The detailed comparison of our dataset with related video datasets is summarized in Tab. 1. We will further process the WebVid-10M dataset and incorporate more filtered data into our dataset.

## A.2 EXPERIMENTAL DETAILS

**Evaluation setting.** To ensure the diversity of the evaluation, each subject in the test set is paired with every bounding box (BBox) during evaluation, and vice versa. This results in a total number

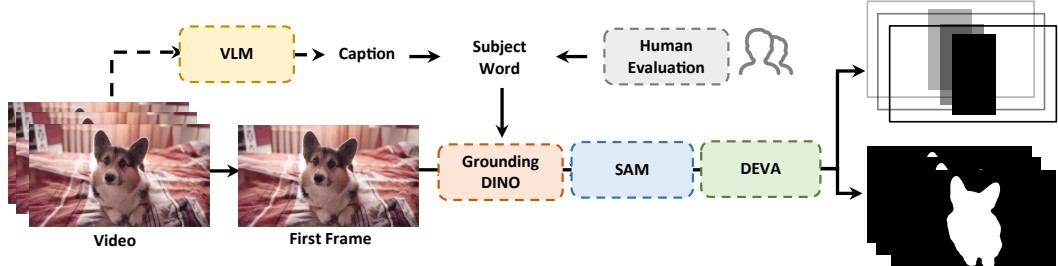

Figure 9: **Pipeline of dataset construction**.

of subject-BBox pairs equal to the product of the number of subjects and bounding boxes, which can fully validate the effectiveness and generalization of our method against baselines. For joint subject customization and motion control, since DreamVideo (Wei et al., 2024) requires reference videos to learn motion patterns and 8 boxes from FreeTraj (Qiu et al., 2024) lack corresponding videos, we solely use 28 bounding boxes from DAVIS videos and 50 subject images, resulting in $50 \times 28 = 1400$ subject-BBox pairs for joint subject customization and motion control. We use all 50 subjects for independent subject customization and all 36 boxes for independent motion control evaluation. For used textual prompts, we design a total of 60 prompt templates, such as "a { } is running on the grass." For a comprehensive assessment, each subject-BBox pair is matched with a randomly selected prompt by replacing "{ }" with the corresponding subject class word.

**Baselines.** Since ModelScopeT2V (Wang et al., 2023a) generates videos at a resolution of 256×256 and exhibits relatively low quality, we adopt ZeroScope, which is further trained on ModelScopeT2V with additional data to produce relatively high-quality videos at a resolution of 576×320, as the base model for all baselines except VideoBooth (Jiang et al., 2024) and MotionCtrl (Wang et al., 2024e), which utilize their collected datasets to train their own models. We follow the default hyperparameter settings from baseline papers for all comparison experiments.

For the task of simultaneously controlling subject appearances and motions, there are currently two methods for us to compare: DreamVideo (Wei et al., 2024) and MotionBooth (Wu et al., 2024a), both requiring fine-tuning at inference time. Since DreamVideo takes reference videos instead of bounding boxes as motion control signals, we use the video corresponding to the bounding boxes from the DAVIS (Pont-Tuset et al., 2017) dataset for training DreamVideo's motion adapter.

In addition, we evaluate the performance of independent subject customization or motion control. For subject customization, we compare our method to DreamVideo and VideoBooth. Since Video-Booth is also a tuning-free framework, we train our DreamCustomizer without the motion encoder and blended mask mechanism, using the same dataset as VideoBooth for a fair comparison. For motion control, we compare our approach with Peekaboo (Jain et al., 2024), Direct-a-Video Yang et al. (2024) and MotionCtrl (Wang et al., 2024e). Both Peekaboo and Direct-a-Video are training-free methods, while MotionCtrl samples 243,000 videos from the WebVid dataset to train its object motion control module. Since MotionCtrl has not yet open-sourced its dataset, we randomly sampled the same number of WebVid videos from our constructed dataset during training for a fair comparison. Here, we only train the motion encoder in our DreamCustomizer to enable motion control.

**Evaluation metrics.** We detail the use of 9 metrics mentioned in the main paper as follows: **1)** For overall consistency, we employ CLIP image-text similarity (CLIP-T), Temporal Consistency (T. Cons.) (Esser et al., 2023), and Dynamic Degree (DD) (Huang et al., 2024) metrics. CLIP-T calculates the average cosine similarity between CLIP (Radford et al., 2021) image embeddings of all generated frames and their text embedding. T. Cons. computes the average cosine similarity across all pairs of consecutive generated frames. DD uses optical flow to measure the motion intensity, following VBench (Huang et al., 2024). **2)** For subject fidelity, we introduce four metrics: CLIP image similarity (CLIP-I), DINO image similarity (DINO-I), region CLIP-I (R-CLIP), and region DINO-I (R-DINO) metrics (Ruiz et al., 2023; Wei et al., 2024; Wu et al., 2024a). CLIP-I and DINO-I use the CLIP model and ViTS/16 DINO Caron et al. (2021) model to compute the average cosine similarities between the subject image and generated frames, respectively. Furthermore, since we focus on subjects appearing in desired positions, we adopt R-CLIP and R-DINO metrics to evaluate the

region subject fidelity, following (Wu et al., 2024a). R-CLIP and R-DINO compute the similarities between the subject image and frame regions defined by bounding boxes. **3)** For motion control precision, we use the Mean Intersection of Union (mIoU) and Centroid Distance (CD) metrics (Qiu et al., 2024). mIoU calculates the average overlap between predicted and ground truth bounding boxes. CD computes the normalized distance between the centroid of the generated subject and target bounding boxes.

### A.3 MORE ABLATION STUDIES

**Effects of reweighted diffusion loss weight $\lambda_{\mathcal{L}}$.** To evaluate the effects of reweighted diffusion loss weight on performance, we test various values of $\lambda_{\mathcal{L}}$, as summarized in Tab. 6. Our results indicate that without using reweighted diffusion loss (*i.e.*, $\lambda_{\mathcal{L}}$=1) results in the poorest performance across most metrics. Increasing $\lambda_{\mathcal{L}}$ to 1.5 or 2 yields improvements in all metrics, confirming that enhancing the loss weight of regions inside bounding boxes during training strengthens subject identity. On the other hand, setting $\lambda_{\mathcal{L}}$ too high (*e.g.*, $\lambda_{\mathcal{L}} = 4$) does not improve subject fidelity metrics but negatively affects motion control metrics such as mIoU and CD. Therefore, we select $\lambda_{\mathcal{L}} = 2$ for our training.

| $\lambda_{\mathcal{L}}$ | CLIP-T | R-CLIP | R-DINO | CLIP-I | DINO-I | T. Cons. | mIoU | CD $\downarrow$ |
|---|---|---|---|---|---|---|---|---|
| 1 | 0.300 | 0.740 | 0.362 | 0.673 | 0.382 | 0.961 | 0.650 | 0.053 |
| 1.5 | 0.302 | 0.745 | 0.370 | 0.687 | 0.385 | 0.965 | **0.676** | 0.050 |
| 2 | **0.303** | **0.751** | **0.392** | **0.694** | **0.411** | **0.968** | 0.670 | **0.048** |
| 4 | 0.298 | 0.750 | 0.389 | 0.693 | 0.399 | 0.964 | 0.647 | 0.056 |

Table 6: **Ablation study on reweighted diffusion loss weight $\lambda_{\mathcal{L}}$.**

### A.4 MORE RESULTS

**Details about the user study.** We conduct a user study involving 20 subjects and 15 motion trajectories, generating 300 videos per method using randomly selected textual prompts. Participants are presented with four sets of questions for each of the three anonymous methods, paired with one reference image and one bounding box sequence indicating motion trajectory. Given the three generated videos in each group, we ask each participant the following questions: (1) Text Alignment: "Which video better matches the text description?"; (2) Subject Fidelity: "Which video's subject is more similar to the target subject?"; (3) Motion Alignment: "Which video's subject movement is more consistent with the target trajectory?"; and (4) Overall Quality: "Which video exhibits better quality and minimal flicker?". Results of the user study are illustrated in Fig. 7.

**More qualitative results.** We showcase more results of joint subject customization and motion control in Fig. 11, providing further evidence of the superiority of our DreamCustomizer.

**Results on Flow Error metric.** To further evaluate the motion control performance, we adopt the Flow Error metric, used by Direct-a-Video, to independently measure the accuracy of subject motion. Specifically, following Direct-a-Video, we compute the Flow Error by (i) calculating frame-wise optical flows for both the generated video and the ground truth video (*i.e.*, the video corresponding to the bounding boxes), (ii) extracting optical flows within the bounding box areas for both videos and (iii) computing the average endpoint error between them. Here, we employ VideoFlow (Shi et al., 2023) to extract optical flow maps. The results are shown in Tab. 7. Our method achieves the best Flow Error, further demonstrating the effectiveness of our motion trajectory control.

| | DreamVideo | MotionBooth | DreamCustomizer |
|---|---|---|---|
| Flow Error $\downarrow$ | 3.717 | 3.710 | **3.158** |

Table 7: **Quantitative comparison on the Flow Error metric.**

**Better qualitative results based on VideoCrafter2.** To further validate the effectiveness of our method and generate more high-quality videos, we retrain our DreamCustomizer on a more powerful

video base model, VideoCrafter2 (Chen et al., 2024b). The generated video resolution is $512 \times 320$ with a fps 8. The frame number is 16. The training setting is the same as our default setting of DreamCustomizer. For inference, we set classifier-free guidance as 12. The fps condition is set to 4. The other inference setting is the same as our default setting.

As illustrated in Fig. 12, the additional results indicate that replacing the backbone with VideoCrafter2 significantly improves video quality, encompassing both aesthetics and clarity. Consequently, this change enhances the transferability and generalization of our method across different models. In fact, our DreamCustomizer represents a novel zero-shot video customization paradigm, and we anticipate that it will function independently of specific foundational models. We also believe that our method could yield even better results when applied to more powerful models.

We present more visual results based on VideoCrafter2 in Fig. 13. We observe that the generated videos exhibit higher quality and natural motion.

A.5   LIMITATIONS AND FUTURE WORKS

In addition to the limitations mentioned in Sec. 6, we also provide several failure cases in Fig. 10. Since we freeze the original 3D UNet parameters during training, our approach is limited by the base model's inherent capabilities, and may fail to generate some rare motions that the subject is unlikely to exhibit. For example, in Fig. 10(a), the basic model fails to generate a video like "a dog is playing guitar on Mars", causing our method to inherit this limitation. Employing more advanced T2V models could mitigate this issue. Another limitation is that our method struggles with decoupling camera and object motion control. As shown in Fig. 10(b), the model may generate videos with moving cameras and static subjects. We propose two solutions to address this issue: (1) Utilize text prompts to control a fixed camera movement, as shown in Fig. 14. Benefiting from the capabilities of pre-trained models, we empirically observed that some prompts, such as "Fixed camera view," can control the static camera movement and alleviate this problem. (2) Construct a dataset with a decoupled camera and object motion using both automated and manual annotation techniques and designing separate modules to control each aspect independently   (Wang et al., 2024e; Yang et al., 2024; Li et al., 2024b).

Future work will focus on overcoming these limitations by leveraging a more powerful base T2V model and separating camera movement from our training dataset. We believe that our proposed method could offer benefits for various real-world applications, including personalized filmmaking, advertising creation, and personal blogging, and inspire future work in customized video generation, such as exploring a unified module for controlling both subject appearance and motion.

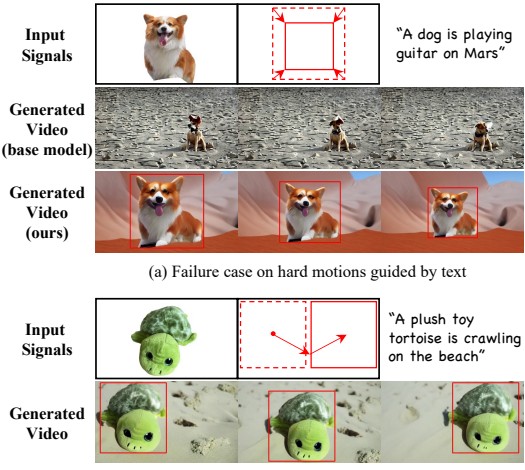

(a) Failure case on hard motions guided by text

(b) Failure case on decoupling camera and motion control

Figure 10: **Failure cases.** (a) Our method is limited by the base model's inherent capabilities. (b) Our method struggles to decouple the camera and motion control.

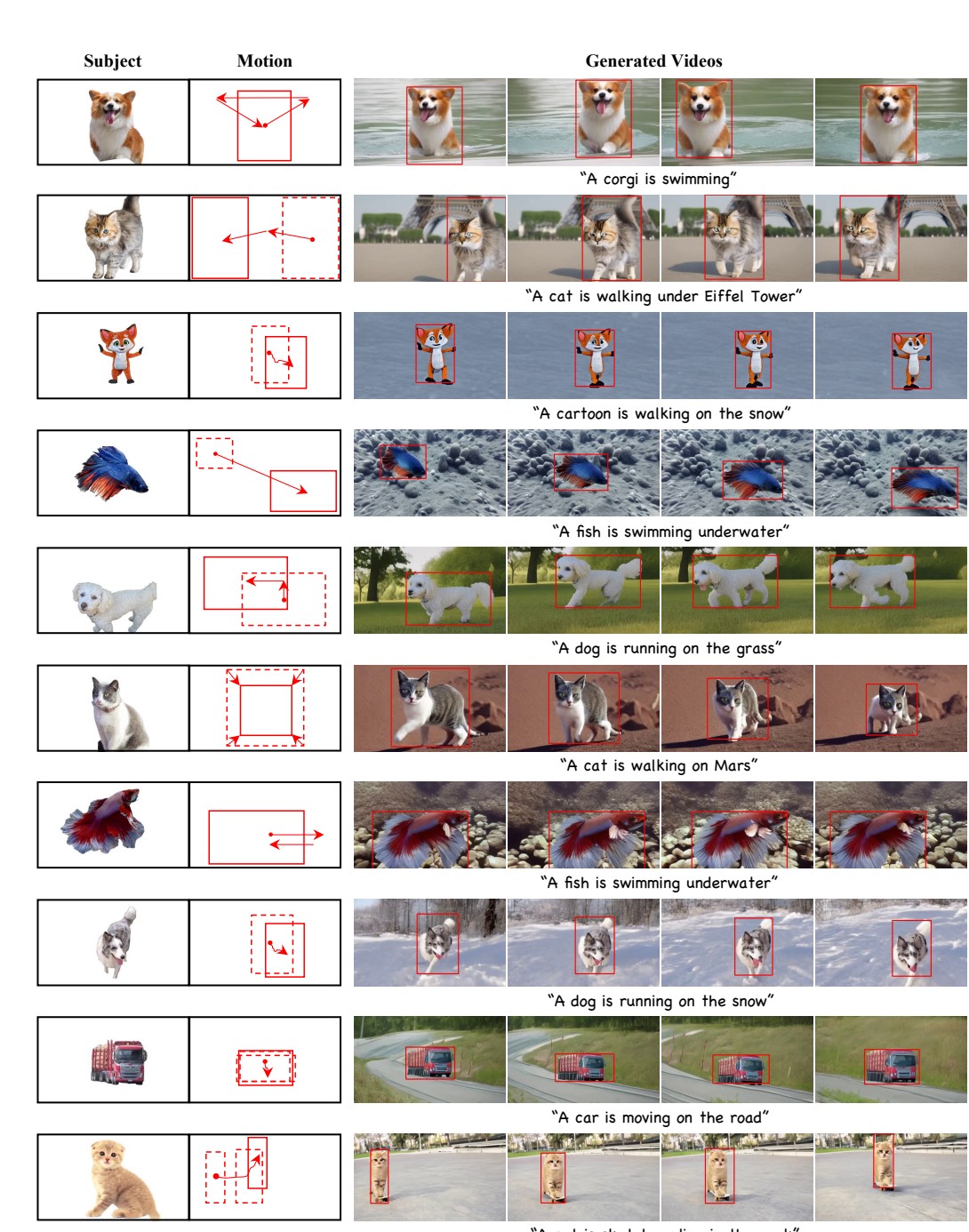

Figure 11: **More qualitative results of DreamCustomizer.**Zoom-in for better view.

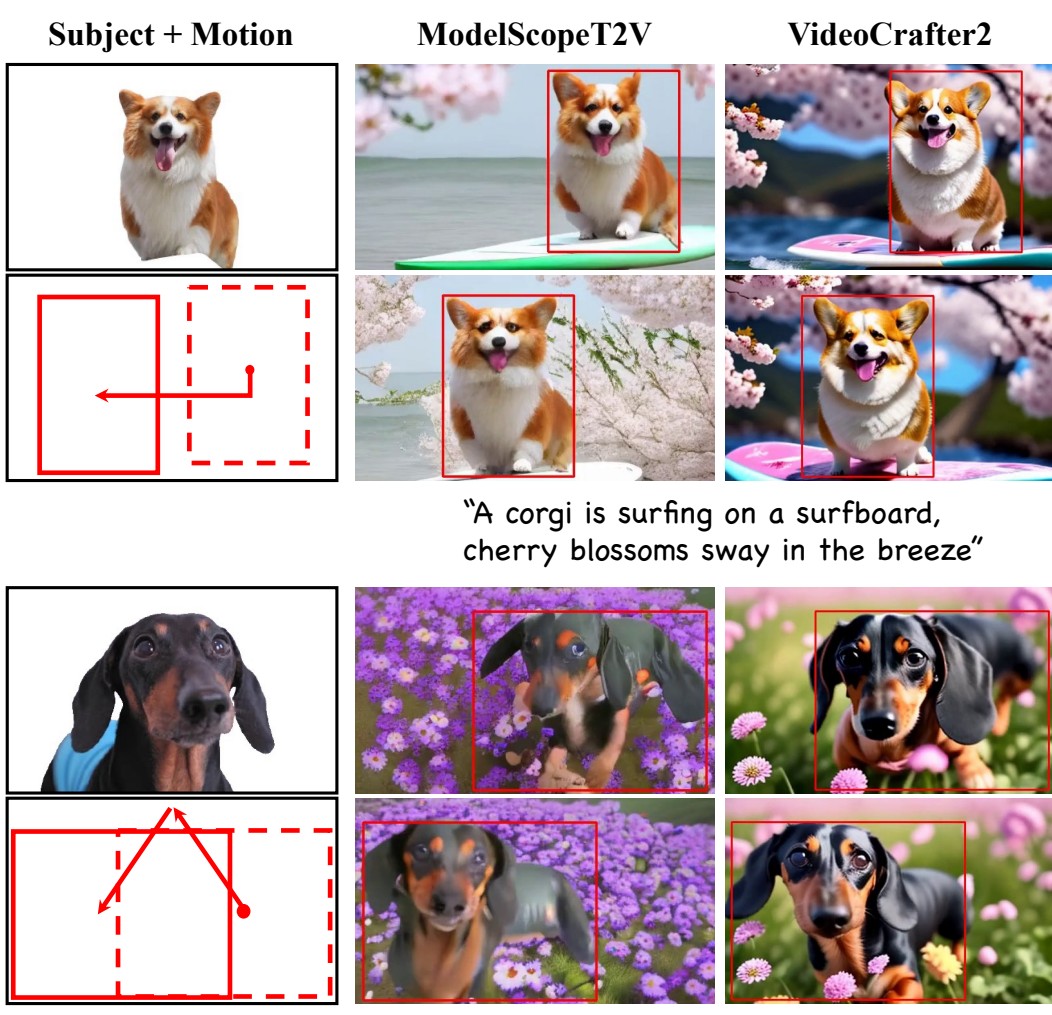

Figure 12: **Video quality comparison between results based on ModelScopeT2V and VideoCrafter2.** Using a more powerful video base model could significantly enhance the generated video quality of our DreamCustomizer.

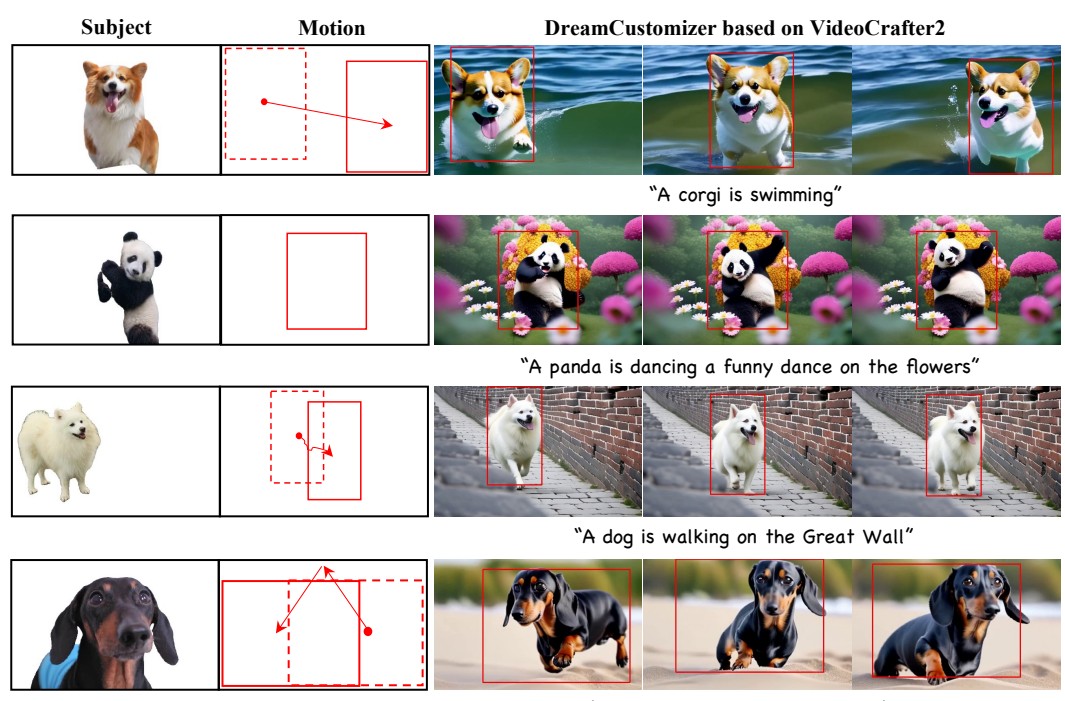

Figure 13: **More qualitative results based on VideoCrafter2.**

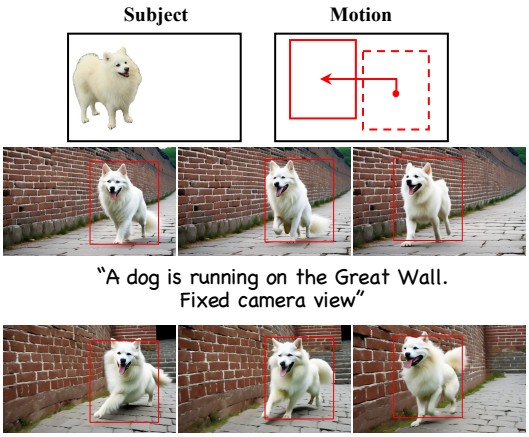

Figure 14: One solution to mitigate the problem of coupling camera movement and subject motion is to utilize textual prompts like "Fixed camera view."

