# OpenReview forum: "Zero-Shot Subject-Driven Video Customization with Precise Motion Control"
_ICLR.cc/2025/Conference — Submitted to ICLR 2025_

### Official Review · Reviewer_XVgX · 2024-10-21

**Soundness:** 4
**Presentation:** 4
**Contribution:** 4
**Rating:** 6
**Confidence:** 5

**Summary:**

The paper introduces a video customization framework, DreamCustomizer, which can generate videos based on a specific subject and motion trajectory.
The framework incorporates two primary components: reference attention for subject learning and a mask-guided motion module for precise motion control.
Experiments demonstrate that DreamCustomizer outperforms state-of-the-art methods in both subject customization and motion control.

**Strengths:**

- The paper is well-written and the figures are properly plotted. Both of them make readers easy to understand the proposed method.

- The experiments are extensive, covering the quantitative and qualitative comparisons with the state-of-the-arts, human evaluation, testing under different conditioning scenarios, and ablation study.

- The authors provide a lot of generated videos, which can help the reader better understand the quality and the potential problems of the proposed method.

- The model is an optimization-free method, which does not require fine-tuning during inference stage.

- The motion control of the proposed model is precise and impressive.

**Weaknesses:**

- Low visual quality:
    - The generated videos have significant artifacts. Did the authors try a better and more recent base model, such as VideoCrafter2?
    - With this video quality, it is hard to judge the fidelity of subject customization.

- Weakness of reconstruction-based method:
    - The model adopts a reconstruction-based training strategy, which collects the input conditions (reference image and bounding boxes) from the video and learn to reconstruct the video with these inputs.
    - However, such training strategy is well-known by worse performance when the user inputs prompt and image from different sources.
    - For example, in the "a corgi is swimming" sample, the model seems to directly copy and paste the input standing corgi and make it floating on the water without introducing reasonable pose or appearance change.
    - Have the authors tried to solve this problem? Applying some image augmentations could alleviate the problem.

**Questions:**

See weaknesses

---

> ### Author Response · Authors · 2024-11-20
> **Responses to Reviewer XVgX**
>
> Thank you for your acknowledgment and insightful comments. We address your concerns as follows.
>
> Notations: W: Weakness
>
> **W-1:**
> **[Video quality]**
> As discussed in Appendix A.5, our method is limited by the inherent capabilities of the base model.
>
> Following your suggestion, we are currently attempting to train our method on VideoCrafter2 to generate more high-quality videos.
> Since VideoCrafter2’s training code is not open-sourced, we have reproduced its training process based on its inference code and the descriptions provided in the original paper.
> We have empirically observed that training on a more robust base model could produce higher-quality videos.
> Due to time and resource constraints, we will strive to present additional visual results during the rebuttal period; otherwise, we promise to showcase them in the final version.
>
> Furthermore, as shown in Figs. 4 and 5, and Tabs. 2 and 3, both qualitative and quantitative results of subject customization demonstrate that our method can capture detailed subject appearances more precisely than SOTA methods.
>
> **W-2:**
> **[Usage of data augmentation]**
> We agree with you that employing data augmentation can help reduce the likelihood of generating videos that resemble a direct copy-paste of subjects.
>
> In fact, we have already used data augmentation to alleviate this issue.
> Specifically, we randomly extract a frame from the entire video as the subject image, which typically differs from the training frames in pose, position, *etc*.
>
> Additionally, many of the videos generated by our method have plausible motions and poses, such as the dancing toy in Fig. 1 and the walking wolf in Fig. 4.
>
> We believe that using more data augmentation techniques, such as color jitter, brightness, and rotation, could further mitigate the mentioned problem.
>
> **[Improvements on reconstruction-based method]**
> We agree with your comment about the limitations of reconstruction-based methods.
> We believe that exploring a non-reconstruction paradigm like Pulid [1] presents a promising research direction, which could seamlessly integrate reward loss to further enhance subject fidelity.
>
> ---
> [1] Guo, Zinan, et al. Pulid: Pure and lightning id customization via contrastive alignment.

---

> ### Comment · Reviewer_XVgX · 2024-11-25
> **Official Comment by Reviewer Qgjf**
>
> Thank you to the authors for their thoughtful feedback. I have carefully reviewed the comments from the other reviewers as well as the authors' responses. In my view, the authors have addressed most of the concerns raised by the reviewers.
>
> However, as noted in my initial comments, the video quality remains a significant issue. This concern was also highlighted by reviewer szvT. While the authors suggest that switching the video backbone to VideoCrafter2 could improve visual quality, the new video samples have not yet been provided in the paper and I cannot check the updated video quality.
>
> Given this, I am revising my score to 6 and leaving the final decision to the area chair.

---

> ### Author Response · Authors · 2024-11-28
> **Follow Up Responses about Video Quality**
>
> Dear Reviewer XVgX,
>
> We would like to express our sincere gratitude for your thorough review and valuable feedback on our method. Your insights have been instrumental in helping us refine and enhance our work.
>
> We first apologize for our delayed response regarding your concerns about the results using VideoCrafter2. The main reason is that VideoCrafter2 does not open-source its training code, so we have been working hard to reproduce its results before transitioning to DreamCustomizer. This process has indeed taken us quite a bit of time, and we apologize once again.
>
> We have trained a preliminary version based on VideoCrafter2, and presented the generated results in Figs. 12 and 13 of our revised paper. The corresponding videos are provided in the supplementary materials. We kindly invite you to review these videos located in `.rebuttal_videos/compare_MS_VC2` and `.rebuttal_videos/results_videocrafter2`.
> As previously discussed, we empirically observe the improved video quality when our method is trained on an advanced video model.
>
> For more analysis of the results, we sincerely invite you to refer to our global response titled "Better Results based on VideoCrafter2".
>
> Once again, we thank you for your careful reading and constructive feedback, which have been crucial in demonstrating the effectiveness and generalizability of our approach through additional experiments with VideoCrafter2.
>
> Best regards,
>
> Authors of paper 249

---

> ### Author Response · Authors · 2024-12-01
> **Looking Forward to Your Response!**
>
> Dear Reviewer XVgX,
>
> We sincerely appreciate your valuable time and thoughtful feedback on our work, and we apologize once again for the delay in our response. We have conducted new experiments based on VideoCrafter2 and incorporated the results into the revised paper and supplementary materials. We kindly invite you to review our paper again in light of these improvements.
>
> We are truly grateful for your recognition of our work. If we have adequately addressed your concerns, we would deeply appreciate it if you could consider reevaluating the score to your initial score. This would mean a great deal to us. Your encouragement and constructive feedback will continue to motivate us to make more substantial contributions to the field of video generation.
>
> Thank you and best regards,
>
> Authors of Paper 249

---

> > ### Author Response · Authors · 2024-12-03
> > **Looking Forward to Your Follow-Up Reply!**
> >
> > Dear Reviewer XVgX,
> >
> > Thank you once again for your valuable time and insightful feedback throughout the review process. As the response period enters its final 8 hours, we are eager to know whether our additional results and clarifications have adequately addressed your concerns. We would sincerely appreciate any further perspectives or discussions you might have at this stage and are more than happy to provide additional clarifications promptly.
> >
> > We greatly appreciate your time and effort. Please let us know if there is anything more we can do to assist.
> >
> > Thank you and best regards,
> >
> > Authors of Paper 249

---

### Official Review · Reviewer_szvT · 2024-10-25

**Soundness:** 3
**Presentation:** 3
**Contribution:** 1
**Rating:** 5
**Confidence:** 4

**Summary:**

The paper introduces DreamCustomizer to generate video with precise motion control and specific subject without the need for fine-tuning during inference. DreamCustomizer leverages "reference attention" and a "mask-guided motion module" to achieve accurate video customizations, controlled by a single subject image and bounding box sequences. The methodology uses masked reference attention and a reweighted diffusion loss to balance subject learning and motion control. The paper claims superior performance over state-of-the-art methods by a new dataset and extensive quantitative and qualitative evaluations.

**Strengths:**

1. The paper distinguishes and mitigates the challenge of motion control dominance by introducing masked reference attention and reweighted diffusion loss, successfully balancing subject fidelity with motion accuracy.

2. The newly curated, diverse dataset provides comprehensive annotations, facilitating training and evaluation for subject and motion control and supporting future research in video customization.

3. With extensive quantitative and qualitative evaluations, the paper demonstrates the effectiveness of DreamCustomizer against state-of-the-art methods.

**Weaknesses:**

1. While DreamCustomizer introduces elements like reference attention and reweighted diffusion loss, these techniques lack substantial novelty. Reference attention, for instance, has been well-studied in prior works such as "StoryDiffusion: Consistent Self-Attention for Long-Range Image and Video Generation," where it has been used to effectively maintain subject consistency. Similarly, the reweighted diffusion loss does not introduce a novel approach to balancing subject fidelity with motion control, as similar weighting techniques have been explored in generative models.

2. DreamCustomizer requires bounding boxes as inputs for motion guidance, similar to control techniques seen in previous works like MotionBooth. This limitation reduces its capability for more complex and high quality video generation tasks and the proposed method is incremental to resolve this problem.

3. Moreover, the presented video quality is not convincing to demonstrate that these proposed components are instrumental in advancing controllable video generation.

**Questions:**

See Weaknesses

---

> ### Author Response · Authors · 2024-11-20
> **Responses to Reviewer szvT**
>
> Thank you for your valuable comments. We address your concerns as follows.
>
> Notations: W: Weakness
>
> **W-1:**
> We study a challenging yet meaningful task: zero-shot video customization with both subject customization and motion control.
> To the best of our knowledge, our framework is the first to achieve this goal in a zero-shot manner, facilitating real-world applications.
> Furthermore, we identify the problem of motion control dominance and address it through our proposed key designs.
> Here, we address your concerns as follows:
>
> **[Differences from Story Diffusion]**
> - Story Diffusion modifies the original self-attention. In contrast, our Reference Attention introduces an additional cross-attention.
> This additional attention is attached to the original self-attention and cross-attention without altering the original self-attention.
> - Our Reference Attention is implemented as a residual structure, which supplements the subject features from the subject image to the original self-attention or cross-attention, a capability absent in Story Diffusion.
> - Story Diffusion randomly samples a limited number of tokens from batch images and concatenates them with the original input tokens. In contrast, our Reference Attention incorporates features of the entire subject image into the video attention features, providing a more comprehensive feature integration strategy between image and video data.
>
> **[Reweighted diffusion loss]**
> We propose a reweighted diffusion loss to address the challenge of motion control dominance in the zero-shot video customization task, explicitly leveraging bounding boxes to differentiate the contributions of regions inside and outside the
> bounding boxes to the standard diffusion loss.
> We believe this design provides valuable insights for future video customization research aimed at balancing potential conflicts among multiple conditions.
>
> **[Other innovations]**
> In addition to reference attention and reweighted diffusion loss, our technical contributions include:
> - The development of a blended mask modeling strategy, along with a masked reference attention mechanism, to enhance subject identity representations at desired positions in the feature level.
> - The introduction of a mask-guided motion module that employs binary box masks as a robust motion control signal, significantly enhancing both motion control precision and training efficiency.
> - The construction of a large, comprehensive, and diverse video dataset to support the zero-shot video customization task.
>
> **W-2:**
> **[Bounding boxes for motion control]**
> We adopt bounding boxes as inputs because they are easier to obtain, more convenient, and more user-friendly compared to other signals.
> Different from previous works, we propose to utilize binary box masks as a robust motion control signal and devise a mask-guided motion module to achieve precise motion control.
>
> In addition, our method demonstrates enhanced control precision compared to previous methods and facilitates complex motion control, such as the contraction and triangular trajectory in Figs. 1 and 11, and the movement from far to near in Fig. 4, which are difficult to achieve by MotionBooth.
>
> To further advance the task of motion control in video generation, a promising approach could leverage the capabilities of Large Language Models (LLMs) to generate intricate trajectories and bounding boxes with greater flexibility.
>
> **W-3:**
> **[Video quality]**
> - Our approach is limited by the inherent capabilities of the base model, as discussed in Appendix A.5.
> Consistent with many previous works, such as Peekaboo, Direct-a-Video, and DreamVideo, we adopt ModelScopeT2V as our base model, which generates videos with a resolution of 256x256 and exhibits relatively low quality.
> In our work, we focus on validating the effectiveness of our method for simultaneously controlling the subject appearance and motion trajectory.
> We believe using a more powerful video base model can improve video quality.
> Furthermore, both qualitative and quantitative results of motion control in Fig. 6 and Tab. 4 demonstrate that our method significantly outperforms the SOTA methods.
> - We have reproduced the training process of VideoCrafter2, a more powerful video diffusion model, and are retraining our method based on it to generate higher-quality videos. Due to time and resource constraints, we will strive to present additional visual results during the rebuttal period; otherwise, we promise to showcase them in the final version.

---

> ### Author Response · Authors · 2024-11-28
> **Follow Up Responses to Reviewer szvT**
>
> Dear Reviewer szvT,
>
> We deeply appreciate your thoughtful feedback and the time you have dedicated to reviewing our work. Your insights are valuable in helping us improve our paper.
>
> Regarding the third weakness you mentioned, we have retrained our method using the more advanced VideoCrafter2 model. We empirically observe that this has led to a further improvement in the quality of the generated videos.
>
> We have included these generated results in Figs. 12 and 13 of the revised paper. Additionally, the corresponding videos are available in the supplementary materials at `./rebuttal_videos`. For more discussions, we sincerely invite you to refer to our global response titled "Better Results based on VideoCrafter2".
>
> We hope that our responses address your concerns, and we look forward to your continued feedback.
>
> Best regards,
>
> Authors of paper 249

---

> > ### Author Response · Authors · 2024-12-01
> > **Looking Forward to Your Response!**
> >
> > Dear Reviewer szvT,
> >
> > We sincerely appreciate your valuable feedback and the time you have dedicated to reviewing our work. Your insights have been instrumental in helping us refine and improve our submission. We have carefully addressed your comments and clarified potential misunderstandings. Additionally, we have provided new experimental results based on VideoCrafter2 with higher-quality videos.
> >
> > We would like to kindly invite you to continue the discussion regarding our work. If you have any additional comments or concerns, please do not hesitate to let us know, and we will do our utmost to address them promptly.
> >
> > Thank you once again for your thoughtful review.
> >
> > Thank you and best regards,
> >
> > Authors of Paper 249

---

> > > ### Author Response · Authors · 2024-12-03
> > > **Looking Forward to Your Follow-Up Reply!**
> > >
> > > Dear Reviewer szvT,
> > >
> > > Thank you once again for your valuable time and insightful feedback throughout the review process. As the response period enters its final 8 hours, we are eager to know whether our additional results and clarifications have adequately addressed your concerns. We would sincerely appreciate any further perspectives or discussions you might have at this stage and are more than happy to provide additional clarifications promptly.
> > >
> > > We greatly appreciate your time and effort. Please let us know if there is anything more we can do to assist.
> > >
> > > Thank you and best regards,
> > >
> > > Authors of Paper 249

---

### Official Review · Reviewer_b9Q7 · 2024-10-29

**Soundness:** 3
**Presentation:** 3
**Contribution:** 3
**Rating:** 6
**Confidence:** 4

**Summary:**

The paper presents a zero-shot approach to subject-driven video generation with controlled motion, showing potential for reducing test-time fine-tuning overhead. The reference attention mechanism and mask-guided motion control provide innovations over previous methods by enhancing the fidelity of subject appearance within controlled bounding boxes.

**Strengths:**

- The DreamCustomizer framework proposed in this paper does not require fine-tuning during inference, and can directly customize the target subject and motion trajectory in a zero-shot situation.

- This generation framework that does not require fine-tuning improves the efficiency of video generation and is conducive to a wide range of practical applications.

- Users only need to provide the subject image and a set of bounding box sequences to generate a custom video, without the need for complex inference stage debugging.

**Weaknesses:**

- While DreamCustomizer claims tuning-free inference, the paper acknowledges the challenge of decoupling camera movement from object motion, leading to camera drift in certain contexts.

-  DreamCustomizer is designed for single-subject customization and does not handle multi-subject videos, which is highlighted as a limitation but without proposed extensions.

- The effectiveness of motion control relies heavily on the precision of bounding box annotations, making the system vulnerable to errors if box tracking is inconsistent.

- Missing some important previous works:

[1] MotionFollower: Editing video motion via lightweight score-guided diffusion. (2024).

[2] FaceChain-ImagineID: Freely crafting high-fidelity diverse talking faces from disentangled audio. CVPR 2024

[3] MotionEditor: Editing video motion via content-aware diffusion. CVPR 2024

[4] Combo: Co-speech holistic 3D human motion generation and efficient customizable adaptation in harmony.  (2024).

**Questions:**

- The paper mentions data filtering steps, but additional details on the refinement of bounding boxes and control signals could provide readers with greater insight into handling imperfect data in training.

- Given the limitations in complex motions and single-subject restriction, it's recommended that the paper discuss specific applications that could benefit from these features as they currently exist.

---

> ### Author Response · Authors · 2024-11-20
> **Responses to Reviewer b9Q7**
>
> Thank you for your insightful feedback and your time. We address your concerns as follows.
>
> Notations: W: Weakness, Q: Question
>
> **W-1:**
> **[Solutions on decoupling camera movement and subject motion]**
> Decoupling of camera movement and subject motion remains a challenging research area actively explored by many scholars today.
> As explained in Appendix A.5, our current approach does not explicitly differentiate between these two types of motion, primarily due to the challenges in creating datasets with annotations for both distinct motion types.
> To address this issue, we propose two possible strategies that require minimal modifications to our method while achieving decoupling:
> - Use text prompts to control static camera movement, ensuring that only subject motion is prioritized.
> - Further refine our dataset by filtering out videos that exhibit camera movement.
>
> In future work, we will attempt to enlarge our dataset to incorporate camera movement conditions and develop a lightweight module to learn camera control.
>
> **W-2:**
> **[Extensions for multi-subject customization]**
> As discussed in Sec. 6, our paper primarily focuses on zero-shot single-subject customization with motion trajectory control.
> For multi-subject customization, potential approaches include constructing video datasets with multiple subjects, decoupling each subject's representations by defining fine-grained masks, and designing a fusion module to handle complex interactions.
> We will explore these extensions in further work.
>
> **W-3:**
> **[Effectiveness of motion control]**
> - We utilize state-of-the-art object detection and tracking models to obtain precise bounding box annotations.
> To further enhance the accuracy of our annotations, we employ both automated and manual annotation techniques.
> We believe leveraging automated annotations in the era of scaling large pre-trained models (*e.g.*, SAM) is necessary, as the effort required to curate a substantially large dataset through human labor is prohibitively high.
>
> - Through rigorous training on our dataset and comprehensive quantitative and qualitative experiments, we demonstrate the effectiveness and robustness of our method's motion control capacity, meaning that the curation pipeline involving automated annotations can produce a model capable of precise motion control.
> In addition, the results presented in Figs. 4 and 6, and Tabs. 2 and 4 further show that our method significantly outperforms baseline methods in motion trajectory control.
>
> **W-4:**
> **[Important previous works]**
> The four works you mentioned also explore topics like motion control or identity preservation in human video generation.
> We have discussed and cited these four works in the revised version.
>
> **Q-1:**
> **[Details on refining annotations]**
> Our process for refining bounding boxes and control signals involves several meticulous steps:
> *(1)* We use the SOTA object detecting and segmentation models with text input to obtain precise bounding boxes and masks.
> *(2)* We set a relatively high threshold to filter out detections that the model is uncertain about.
> *(3)* We filter out videos that contain multiple subjects for the subject word by the number of bounding boxes in the first frame. We also filter out subjects that are either too small or too large (*i.e.*, those nearly matching the size of the entire video) by assessing the ratio of the width, height, and area of the subject’s bounding box to the entire video.
> *(4)* We use the NLTK library to perform lemmatization and filter out non-words.
> *(5)* We ask human annotators to refine the class words of the desired subjects to ensure that the correct text is fed into the detection model and the expected results are obtained.
>
> To further enhance annotation precision, it may be beneficial to utilize more robust models and involve human evaluation in refining box and mask annotations.
> We have included these detailed descriptions in the revised version.
>
> **Q-2:**
> **[Discussion on specific applications]**
> Our method offers benefits for various real-world applications, including personalized filmmaking, advertising creation, and personal blogging.
> Furthermore, we believe that our approach can inspire future work in customized video generation, such as exploring a unified module for controlling both subject appearance and motion.

---

> > ### Author Response · Authors · 2024-11-28
> > **Follow Up Responses to Reviewer b9Q7**
> >
> > Dear Reviewer b9Q7,
> >
> > We greatly appreciate your time and feedback on our work. Your kind suggestions have been valuable in helping us clarify and improve our work.
> >
> > We would like to elaborate further on our previous response regarding **[Solutions on decoupling camera movement and subject motion]**. Regarding our mention of "Using text prompts to control static camera movement," we have included examples in Fig. 14 of the revised paper to clarify this point. We kindly invite you to review the relevant videos in the supplementary materials at `./rebuttal_videos/decouple_motion`.
> >
> > We hope that our responses adequately address your concerns, and we look forward to your feedback.
> >
> > Best regards,
> >
> > Authors of paper 249

---

> > > ### Author Response · Authors · 2024-12-01
> > > **Looking Forward to Your Response!**
> > >
> > > Dear Reviewer b9Q7,
> > >
> > > We greatly appreciate your recognition of our work and your insightful feedback, which have significantly contributed to the clarity and enhancement of our work. We have carefully addressed your concerns, provided more detailed clarifications, supplemented our paper with new experimental results concerning the decoupling of camera movement and subject motion, and cited relevant excellent work in the field.
> > >
> > > We kindly invite you to revisit our paper in light of these updates and clarifications. We sincerely hope our responses thoroughly address your concerns.
> > >
> > > Thank you once again for your dedicated effort as a reviewer!
> > >
> > > Thank you and best regards,
> > >
> > > Authors of Paper 249

---

> > > > ### Author Response · Authors · 2024-12-03
> > > > **Looking Forward to Your Follow-Up Reply!**
> > > >
> > > > Dear Reviewer b9Q7,
> > > >
> > > > Thank you once again for your valuable time and insightful feedback throughout the review process. As the response period enters its final 8 hours, we are eager to know whether our additional results and clarifications have adequately addressed your concerns. We would sincerely appreciate any further perspectives or discussions you might have at this stage and are more than happy to provide additional clarifications promptly.
> > > >
> > > > We greatly appreciate your time and effort. Please let us know if there is anything more we can do to assist.
> > > >
> > > > Thank you and best regards,
> > > >
> > > > Authors of Paper 249

---

### Official Review · Reviewer_gEDD · 2024-11-01

**Soundness:** 3
**Presentation:** 2
**Contribution:** 2
**Rating:** 5
**Confidence:** 4

**Summary:**

The paper introduces DreamCustomizer, a video customization model capable of generating videos with a specific subject and motion trajectory. Technically, DreamCustomizer incorporates reference attention into the T2V model to learn the appearance information of the subject, and employs a mask-guided motion module to achieve motion control based on a sequence of bounding boxes. Additionally, the paper proposes masked reference attention and reweighted diffusion loss to emphasize the model's focus on learning from the subject input. Extensive experiments conducted on a newly curated dataset demonstrate the effectiveness of DreamCustomizer for subject customization and motion control.

**Strengths:**

S1: The paper introduces DreamCustomizer, a video generation method capable of controlling both the appearance of the generated subject and the motion trajectory. Additionally, it proposes two strategies, masked reference attention and reweighted diffusion loss, to enhance the model's focus on learning subject appearance.

S2: DreamCustomizer constructs a dataset from WebVid-10M for training, enabling subject customization and motion control during inference without the need for fine-tuning.

S3: The paper presents good qualitative and quantitative results.

**Weaknesses:**

W1: The description of the dataset used for evaluation in Section 5.1 could benefit from further clarification. I would appreciate it if the authors could specify the total number of subject-BBox pairs. Additionally, the statement "we design 60 textual prompts for validation" raises a question for me: does this imply that there are only 60 pairs used for evaluation? It seems that each input pair would typically have a corresponding text prompt.

W2：I’m unsure if I missed this detail, but I would like to know whether Tables 2, 3, and 4 are evaluated on the same validation set. If so, I’m curious why the values for DreamCustomizer differ for the same metrics, such as CLIP-T and Temporal consistency, across the three tables.

W3：While the qualitative comparison indicates that DreamCustomizer achieves better mIoU and CD, Figure 10 shows that the model often generates camera movement rather than subject movement. This type of camera movement is not the intended form of motion in this paper and typically results in low CD values. My concern is that the metrics CD and mIoU may not accurately assess the appropriateness of the generated video’s motion (e.g., distinguishing between camera movement and subject movement), potentially reducing the credibility of these metrics.

**Questions:**

Q1: I hope the authors can address the concerns raised in the "Weaknesses" section.

---

> ### Author Response · Authors · 2024-11-20
> **Responses to Reviewer gEDD**
>
> Thank you for your valuable comments, and we apologize for not clarifying the experimental settings.
> We address your concerns as follows.
>
> Notations: W: Weakness
>
> **W-1:**
> **[Clarification on subject-BBox pairs]**
> To ensure the diversity of the evaluation, each subject is paired with every bounding box during evaluation, and vice versa. This results in a total number of subject-BBox pairs equal to the product of the number of subjects and the number of bounding boxes, which can fully validate the effectiveness and generalization of our method against baselines.
> For joint subject customization and motion control, since DreamVideo requires reference videos to learn motion patterns and 8 boxes from FreeTraj lack corresponding videos, we solely use 28 bounding boxes from DAVIS videos and 50 subject images, resulting in 50x28=1400 subject-BBox pairs for results in Tab. 2.
> We use all 50 subjects for independent subject customization and all 36 boxes for independent motion control evaluation.
>
> **[Clarification on prompts for validation]**
> We design a total of 60 prompt templates, such as "a { } is running on the grass."
> For a comprehensive assessment, each subject-BBox pair is matched with a randomly selected prompt by replacing "{ }" with the corresponding subject class word.
> Consequently, the results in Tab. 2 are based on a total of 1400 prompts.
> We will make the test set data and all prompt templates publicly available.
>
> **W-2:**
> **[Details about Tabs. 2, 3, and 4]**
> For a fair comparison, Tabs. 2, 3, and 4 are reported on the same validation set (50 subjects and 36 boxes) but pertain to different experimental setups, each with separately trained models.
> Detailed settings for results in Tab. 2 are discussed above in **W-1**.
> Tab. 3 reports the results of independent subject customization without the need for motion control; therefore, we retrain the model with reference attention and reweighted diffusion loss using the VideoBooth dataset to ensure a fair comparison.
> Tab. 4 shows the results of independent motion control without subject learning, so we retrain our model using only the mask-guided motion module.
> More details on the implementation and experiments can be found in Appendix A.2.
> Therefore, the results in Tabs. 2, 3, and 4 vary when evaluated using the same metrics.
>
> **W-3:**
> **[Evaluation on subject motion]**
> To facilitate comparison, we adopt the commonly used CD and mIoU metrics to measure subject motion, following prior work such as Peekaboo and FreeTraj.
> However, we agree with you that existing metrics for evaluating motion control are limited in distinguishing between camera movement and subject motion.
> To address your concerns, we further validate the effectiveness of our method from two aspects:
> - **[Results on a new metric]** We adopt the Flow Error metric, used by Direct-a-Video, to independently measure the accuracy of subject motion. Specifically, following Direct-a-Video, we compute the Flow Error by *(1)* calculating frame-wise optical flows for both the generated video and the ground truth video (*i.e.*, the video corresponding to the bounding boxes), *(2)* extracting optical flows within the bounding box areas for both videos and *(3)* computing the average endpoint error between them.
> Here, we employ VideoFlow [1] to extract optical flow maps.
> The results are shown as follows. Our method achieves the best Flow Error, further demonstrating the effectiveness of our motion trajectory control.
>
>   |                              |      DreamVideo     |      MotionBooth     |      DreamCustomizer(ours)      |
>   |------------------------------|:-------------------:|:--------------------:|:-------------------:|
>   |           **Flow Error ↓**   |        3.717        |        3.710         |       **3.158**     |
>
>
> - **[Results on subject motion videos]** We manually filter out videos that tend to exhibit camera movement and recalculate the mIoU and CD metrics on the remaining subject motion videos.
> For a fair comparison, we evaluate on the intersection of subject motion videos of all methods. The results are shown as follows.
> As the reviewer mentioned, we find that videos with camera movement can result in a lower CD.
> After removing these videos, the CD and mIoU of all methods become worse.
> However, our method still achieves the best performance on subject motion videos, which further validates its effectiveness.
>
>   |                | DreamVideo | MotionBooth | DreamCustomizer(ours) |
>   |:--------------:|:----------:|:-----------:|:---------------:|
>   | **mIoU ↑**     |   0.100    |    0.272    |     **0.551**   |
>   | **CD ↓**       |   0.225    |    0.108    |     **0.064**   |
>
>
> ---
> [1] Shi, Xiaoyu, et al. Videoflow: Exploiting temporal cues for multi-frame optical flow estimation.

---

> > ### Comment · Reviewer_gEDD · 2024-11-26
> >
> > First of all, I really appreciate the detailed reintroduction of the various experimental settings, which addressed my previous doubts regarding the experimental setup. Additionally, to reflect the accuracy of Bbox control, the authors introduced flow error for evaluation. In fact, this should exhibit a trend similar to CD and cannot avoid the issues caused by camera movement.
> >
> > The authors further manually filtered out cases with camera movement. However, the occurrence of camera movement varies across different methods and cases. Could the authors provide the structure of the dataset after filtering? I am concerned that most of the cases involve camera movement, and after filtering, there may not be enough remaining cases to make a meaningful comparison.
> >
> > Finally, based on the video cases provided by the authors, my current concerns can be summarized in two points: first, the quality of the generated videos is relatively low, with most cases showing unnatural motion; second, this method inevitably generates many cases of camera movement rather than reasonable object movement.

---

> ### Author Response · Authors · 2024-11-28
> **Follow Up Responses to Reviewer gEDD**
>
> Dear Reviewer gEDD,
>
> Thank you for your kind comments and valuable feedback. We address your concerns point by point.
>
> > Could the authors provide the structure of the dataset after filtering?
>
> We apologize for any confusion and appreciate the opportunity to clarify.
> After data filtering, our method retain 911 videos, while the DreamVideo method retains 835 and MotionBooth retains 829. To ensure a fair comparison, we identify the intersection of subject motion videos across all three methods, resulting in a subset of 718 videos for evaluation. These 718 videos encompass 50 distinct subjects and 20 different bounding boxes.
> The details about the dataset structure are reported in the following Table 1.
> Therefore, we believe this subset is diverse and sufficient for making a meaningful comparison.
>
> Table 1. Subject motion evaluation dataset
> |                           |   DreamVideo   |   MotionBooth   |   DreamCustomizer   |   Intersection   |
> |---------------------------|:--------------:|:---------------:|:-------------------:|:----------------:|
> | Video number after filtering   |      835       |       829        |        911         |       718        |
> | Subject number after filtering |       50       |        50        |         50         |        50        |
> | Motion number after filtering  |       22       |        21        |         23         |        20        |
>
> > first, the quality of the generated videos is relatively low, with most cases showing unnatural motion
>
> To address your concerns, we invite you to refer to our global response titled "Better Results based on VideoCrafter2". We provide new results in the revised paper in Figs. 12 and 13, with the corresponding videos provided in the supplementary materials, which are located in `./rebuttal_videos`. We empirically observed that using a more powerful model, such as VideoCrafter2, could generate more natural motion.
>
> > second, this method inevitably generates many cases of camera movement rather than reasonable object movement
>
> We appreciate your feedback and sincerely apologize for any confusion caused by the lack of details regarding motion control. We would like to address your concerns by the following three points:
> - According to Table 1, our method effectively generates more videos featuring prominent object motion rather than camera movement. Although excluded from the metric calculation, most of the videos with camera movement still exhibit noticeable object movement. We believe that these slight camera movements do not diminish the saliency of the object motion.
> - Decoupling camera movement from subject motion is a complex task and a key focus in current research. This challenge arises from the intertwined motion patterns in video data, including the intrinsic motion of the subject itself, the subject movement in space, and the camera movement. These coupled motions complicate model training. Previous work, such as MotionCtrl [1], has attempted to address this by creating modules to control camera and subject motion separately. Although MotionCtrl has achieved promising results for independent subject motion control, it still exhibits camera movement in some cases when generating specified object motion, as demonstrated in the video provided in our supplementary material (located in `./rebuttal_videos/decouple_motion/MotionCtrl`). This further underscores the challenge of fully decoupling camera movement and object motion.
> - To address this challenge, we propose two solutions:
>     - Utilize text prompts to control a fixed camera movement. For example, use the text prompt "fixed camera view", as shown in the Fig. 14 of the revised paper. The corresponding videos are located in `./rebuttal_videos/decouple_motion`. The results illustrate that this strategy can mitigate the coupling of camera movement and subject motion.
>     - Construct a dataset with decoupled camera movement and object motion using both automated and manual annotation techniques, and design separate modules to control each aspect independently.
>
> Finally, we sincerely appreciate your great efforts in reviewing this paper once again. Your constructive advice and valuable comments really help improve our paper.
>
> Best regards,
>
> Authors of paper 249
>
> ---
>
> [1] Wang, Zhouxia, et al. "Motionctrl: A unified and flexible motion controller for video generation."

---

> ### Author Response · Authors · 2024-12-01
> **Looking Forward to Your Response!**
>
> Dear Reviewer gEDD,
>
> Thank you once again for your time and effort in reviewing our work. Your feedback has been invaluable in helping us clarify, improve, and refine our paper. We have worked diligently to address your comments.
>
> We would like to invite you to continue the discussion regarding our work. We hope that our responses have successfully addressed your concerns, and we would sincerely appreciate it if you could consider reevaluating your rating.
>
> Thank you and best regards,
>
> Authors of Paper 249

---

> > ### Author Response · Authors · 2024-12-03
> > **Looking Forward to Your Follow-Up Reply!**
> >
> > Dear Reviewer gEDD,
> >
> > Thank you once again for your valuable time and insightful feedback throughout the review process. As the response period enters its final 8 hours, we are eager to know whether our additional results and clarifications have adequately addressed your concerns. We would sincerely appreciate any further perspectives or discussions you might have at this stage and are more than happy to provide additional clarifications promptly.
> >
> > We greatly appreciate your time and effort. Please let us know if there is anything more we can do to assist.
> >
> > Thank you and best regards,
> >
> > Authors of Paper 249

---

### Author Response · Authors · 2024-11-20
**Summary of the Author Rebuttal**

We sincerely thank all the reviewers for their valuable feedback and for recognizing the strengths of our work.
Specifically, we appreciate their acknowledgment of the effectiveness of the components in the proposed DreamCustomizer (Reviewer gEDD), the impressive motion control capacity (Reviewer XVgX), the practicality of the tuning-free approach (Reviewers b9Q7 and XVgX), the construction of a large-scale, diverse dataset (Reviewers gEDD and szvT), and the comprehensive experiments and ablation studies (Reviewers gEDD, szvT and XVgX).

We would like to clarify our key contributions:
1. We propose a tuning-free video customization framework to control both subject appearance and motion trajectory using the devised reference attention and the mask-guided motion module.
2. We identify the problem of motion control dominance and address it by proposing masked reference attention and reweighted diffusion loss.
3. We curate a large, comprehensive, and diverse video dataset to facilitate future research in video customization.

Moreover, we comprehensively address the concerns of reviewers in detail within our subsequent responses, and have revised the manuscript accordingly with changes reflected in blue.

These changes include more details on the evaluation setting and data annotation, added experimental results on subject motion evaluation, extended discussions on applications and future work, and added related works.

---

### Author Response · Authors · 2024-11-28
**Better Results based on VideoCrafter2**

We sincerely appreciate the reviewers’ constructive feedback, which has been invaluable in improving our work. We also express our gratitude for their patience and time in reviewing our paper.

To generate higher quality videos, we have reproduced the training process of VideoCrafter2, a recent advanced model, and trained a preliminary version based on VideoCrafter2 with a larger resolution of 512x320, compared to the 448x256 resolution of our trained ModelScopeT2V.

The generated results are included in the revised paper in Appendix A.4 and A.5, and the corresponding videos have been added to the supplementary materials. We kindly invite the reviewers to review the videos located in `./rebuttal_videos`, which should address the concerns regarding the video quality.

The visual results in Fig. 12 of the revised paper indicate that replacing the backbone with VideoCrafter2 further improves video quality, encompassing both aesthetics and clarity. We also provide additional visual results based on VideoCrafter2 in Fig. 13 to showcase more generated high-quality videos.

These experimental results validate the transferability and generalization of our method across different models. In fact, our DreamCustomizer represents a novel zero-shot video customization paradigm, and we anticipate that it will function independently of specific foundational models. We also believe that our method could yield even better results when applied to more powerful models.

Once again, we thank all reviewers for their invaluable suggestions, which have greatly strengthened our approach.

---

### Meta-Review · Area_Chair_eh21 · 2024-12-21

**Metareview:**

This work presents DreamCustomizer, a video generation method capable of controlling both subject appearance and motion trajectory in a tuning-free manner. Despite the authors' efforts to address the reviewers' concerns through clarifications and additional experiments, the issues related to limited novelty, low visual quality generation and limited motion control remain unresolved. Therefore, this paper, in its current form, does not meet the standards for acceptance on ICLR. The authors are encouraged to address the mentioned limitations and further strengthen this work.

**Additional Comments On Reviewer Discussion:**

Most reviewers engaged in discussions with the authors except for Reviewer b9Q7. However, most reviewers still have concerns about novelty, video generation quality and sufficiency of experiments.

---

### Decision · Program_Chairs · 2025-01-22

Reject